# Enhanced CD95 and interleukin 18 signalling accompany T cell receptor Vβ21.3+ activation in multi-inflammatory syndrome in children

Zhenguang Zhang [1], Iain R. L. Kean [1], Lisa M. Dratva [2], John A. Clark [1], Eleni Syrimi[3], Naeem Khan[3], Esther Daubney[4], Deborah White[4], Lauran O'Neill[5], Catherine Chisholm[5], Caroline Payne[5], Sarah Benkenstein[5], Klaudia Kupiec [5], Rachel Galassini[6], Victoria Wright [6], Helen Winmill[7], Ceri Robbins[7], Katherine Brown[5], Padmanabhan Ramnarayan[6], Barnaby Scholefield [7,8], Mark Peters[5,9], Nigel Klein [5,9], Hugh Montgomery[10], Kerstin B. Meyer [2], Sarah A. Teichmann [2,11], Clare Bryant [12,13] ✉, Graham Taylor [3,13] ✉ & Nazima Pathan [1,4,13] ✉

Multisystem inflammatory syndrome in children is a post-infectious presentation SARS-CoV-2 associated with expansion of the T cell receptor Vβ21.3+ T-cell subgroup. Here we apply muti-single cell omics to compare the inflammatory process in children with acute respiratory COVID-19 and those presenting with non SARS-CoV-2 infections in children. Here we show that in Multi-Inflammatory Syndrome in Children (MIS-C), the natural killer cell and monocyte population demonstrate heightened CD95 (Fas) and Interleuking 18 receptor expression. Additionally, TCR Vβ21.3+ CD4+ T-cells exhibit skewed differentiation towards T helper 1, 17 and regulatory T cells, with increased expression of the co-stimulation receptors ICOS, CD28 and interleukin 18 receptor. We observe no functional evidence for NLRP3 inflammasome pathway overactivation, though MIS-C monocytes show elevated active caspase 8. This, coupled with raised IL18 mRNA expression in CD16- NK cells on single cell RNA sequencing analysis, suggests interleukin 18 and CD95 signalling may trigger activation of TCR Vβ21.3+ T-cells in MIS-C, driven by increased IL-18 production from activated monocytes and CD16- Natural Killer cells.

While severe illness in children following SARS-CoV-2 infection is rare, a spectrum of disease exists in this younger patient cohort. In some children, minor or even asymptomatic SARS-CoV-2 infection is followed some weeks later by a hyper-inflammatory syndrome - multi-system inflammatory syndrome in children (MIS-C). Clinically, the presenting symptoms of MIS-C overlap with those of toxic shock syndrome (TSS) and Kawasaki disease[1]. There has been much speculation on the potential shared immune pathophysiology between these diseases.

There appear to be immunological similarities between MIS-C and TSS. Increased frequencies of TCR Vβ21.3+ (gene name: *TRBV-11-2*) positive T cells are observed in both CD4+ and CD8+ T cells in MIS-C[2–5], which resembles superantigen activation of specific Vβ subsets of T cells (e.g. TSS toxin-1 that targets Vβ–2 T cells)[6,7]. In MIS-C a variety of

cytokines are increased in plasma, including Th1 type IFN-γ and downstream chemokines CXCL9 and CXCL10[2], as seen in TSS[8]. However, while CD4+ T cells have a clear dominant role in TSS development[9], knowledge of CD4+ T cells in MIS-C is limited[10] and T helper (Th) cell differentiation has not been explored. Monocyte transcription is reported to shift towards a sepsis state, using signature genes from bacterial sepsis blood monocytes[11], but functional-level studies are lacking. Several DNA mutations related to the inflammatory response have been reported in a small proportion of MIS-C cases[12–14]. However, for most children with MIS-C, the immunopathogenesis of this post-infection phenomenon remains an enigma, and the signals linking activation between innate and adaptive immune cells are still obscure.

In Kawasaki disease, NLRP3 inflammasome mediated interleukin 1 (IL-1) family cytokines are important drivers of vasculitis activation[15,16], and these cytokines are also increased in MIS-C[17]. The monocyte NLRP3-mediated inflammasome response is key to the innate immune recognition of pathogen- and damage-associated molecular patterns, such as pathogen-associated RNA, bacterial and fungal toxins, particulate matter, and metabolites such as ATP[18,19]. When NLRP3 is activated after priming, it recruits caspase 1 to cleave pro-IL-1β and pro-interleukin 18 (pro-IL-18). It also processes the cell death protein gasdermin D to lyse cells, releasing a range of bioactive danger proteins alongside the active IL-1β and IL-18 cytokines. Despite its pivotal role in inflammation, including in Kawasaki Disease, the activity of the NLRP3 pathway in MIS-C is poorly understood.

Using functional cellular assays of endotoxin and NLRP3 responses, alongside high dimensional cytokine, mass cytometry, and scRNA-seq analysis, we studied the immune landscape of MIS-C from acute to convalescence stages, comparing it with severe paediatric COVID-19, other non-SARS-CoV-2 related paediatric infections, and healthy children. We provide a detailed characterisation of the expanded TCR Vβ21.3+ T cells together with NK cells and monocytes in MIS-C. No overactive NLRP3 inflammasome pathway activity could be seen, but increased FAS expression and active caspase 8 activity was seen in the monocytes and it is possible this could act as an alternative processor of inflammasome substrates in MIS-C patients.

## Results
### Clinical characteristics
In this prospective multicentre observational study, we enrolled 43 children with MIS-C, 18 children with acute SARS-CoV-2 infection (requiring admission into paediatric intensive care unit (PICU)), 40 children with non-SARS-CoV-2 related illnesses and 7 healthy children. There were 4 parts of the study: (i) 48-plex biomarker assay of plasma samples, (ii) high dimensional cytometry by time of flight (CyToF) analysis of peripheral blood mononuclear cells (PBMCs) using two complementary cell surface marker panels, (iii) scRNA-Seq analysis, and (iv) whole blood functional analyses (Fig. 1A). In line with previous reports[20–22], we observed several abnormalities in the admission full blood count and biochemistry measurements of MIS-C patients (collected within 48 h of admission) (Fig. 1B). Lymphopenia was found in most children, and monocyte numbers tended to be lower than normal range. Acute inflammatory markers including C-reactive protein (CRP) and ferritin were increased in all children with MIS-C, along with alanine aminotransferase (ALT). Thrombosis marker D-dimer and cardiac injury marker troponin were also increased in children with MIS-C. There was a significant positive correlation between neutrophil and whole blood counts. Lymphocyte count was negatively correlated with D-dimer level. This was also the case for monocyte count and CRP level (Fig. 1C).

The median age of children with MIS-C was 10.96 years (IQR 8.0–13.31 years), and the ethnicity was predominantly white and Asian (42% and 26%), with a slightly higher ratio of males (60%). While the paediatric control group had a similar age range (median: 9.45 years, IQR: 4.45–14.7 years) to the MIS-C cohort, the age of the COVID-19

pneumonia group was lower (median: 3.7 years, IQR: 1.2–12.44 years; summary table of Fig. 1D). In the healthy control group, the average age was 12 years old, with 5 males and 2 females. Further clinical details including MIS-C symptoms and sampling times are provided in Supplementary Data 1.

### Both inflammatory and anti-inflammatory cytokines are increased in MIS-C
Cytokine levels in MIS-C patients have previously been compared to healthy subjects and other paediatric conditions like hemophagocytic lymphohistiocytosis, and acute SARS-CoV-2 infection[11,23]. To our knowledge, there has been no comparison to children with infections requiring intensive care support. We therefore performed a multiplex Luminex assay for 48 soluble protein biomarkers in plasma samples, comparing children with MIS-C to two other PICU patient cohorts: children with acute SARS-CoV-2 infection or acute non-SARS-CoV-2 lower respiratory tract infection (LRTI).

Dimensionality reduction by principal component analysis (PCA) of all 48 cytokines separated MIS-C and control PICU patients with acute SARS-CoV-2 infection and LRTI by dimension 2 (different cytokines, Fig. 2A). Compared to the two PICU infection groups, children with MIS-C had significantly higher levels of multiple cytokines/chemokines. These include Th17-related IL-17A, Th1 cell-related IL-12 p40, and myeloid cell-related TNF-α, IL-1β, and IL-18. Also higher were antiviral type 1 interferon IFN-α2 and type 2 interferon IFN-γ, IFN-γ induced chemokines CXCL9 and CXCL10; soluble IL-2Rα (sCD25, key cell membrane signal conductor for anti-inflammatory T regulatory cells), type 2 inflammatory cytokine IL4, anti-inflammatory cytokine IL-10, IL1RA, growth factors M-CSF and FGF and monocyte chemoattractant factor MCP-3 (Fig. 2B–Q). In contrast, TNF-β levels were significantly higher in children with LRTI (Fig. 2R).

The increased levels of both inflammatory and anti-inflammatory cytokines we and others have observed in MIS-C are also a feature of sepsis. Therefore, we also measured sepsis-related LPS (lipopolysaccharides, also called endotoxin) and LBP (LPS binding protein) levels. Although LBP was raised in MIS-C compared to PICU LRTI, there was no difference in LPS levels (Fig. 2S, T).

IL-18 is a driver of IFN-γ signalling[24]. Plasma IL-18 activity is regulated by the neutralising protein IL-18 binding proteins (IL-18 BP, with IL-18 BPa as the main splice variant). To investigate IL-18 activity, we measured IL-18 BPa to determine the levels of active free IL-18. Although MIS-C patients had elevated IL-18 levels they also had elevated IL-18 BPa levels compared to PICU LRTI patients (Fig. 2U). Taking both measurements into account showed there was no significant difference in calculated free IL-18 levels between MIS-C and LRTI patients (Fig. 2V).

Examining correlations between plasma biomarkers and clinical features for the MIS-C patients revealed cardiomyocyte injury marker troponin was positively correlated with circulating IL-1α and IL-1β levels (Supplementary Fig. 1A, B), supporting the proposed link between IL-1 family cytokines and myocardial injury[25]. Fibrinolysis product D-dimer also positively correlated with these 2 cytokines, as well as MCP-1 (Supplementary Fig. 1C, D). In addition, neutrophil frequency positively correlated with HGF levels (Supplementary Fig. 1E). For the six MIS-C patients for whom serial samples were available, we found soluble IL-2Rα, M-CSF and TNF-α levels were significantly decreased during the treatment course, with a trend towards decreased IFN-γ levels (Supplementary Fig. 1G).

### Upregulation of T cell activation, immune checkpoint and co-stimulation receptor markers in TCR Vβ21.3+ T cells in MIS-C
To better understand immune cell status in MIS-C, we analysed PBMCs using high dimensional mass cytometry with two antibody panels. These panels were designed to analyse a range of cell types but particularly monocytes and T cells (Supplementary Tables 1 and 2). Given

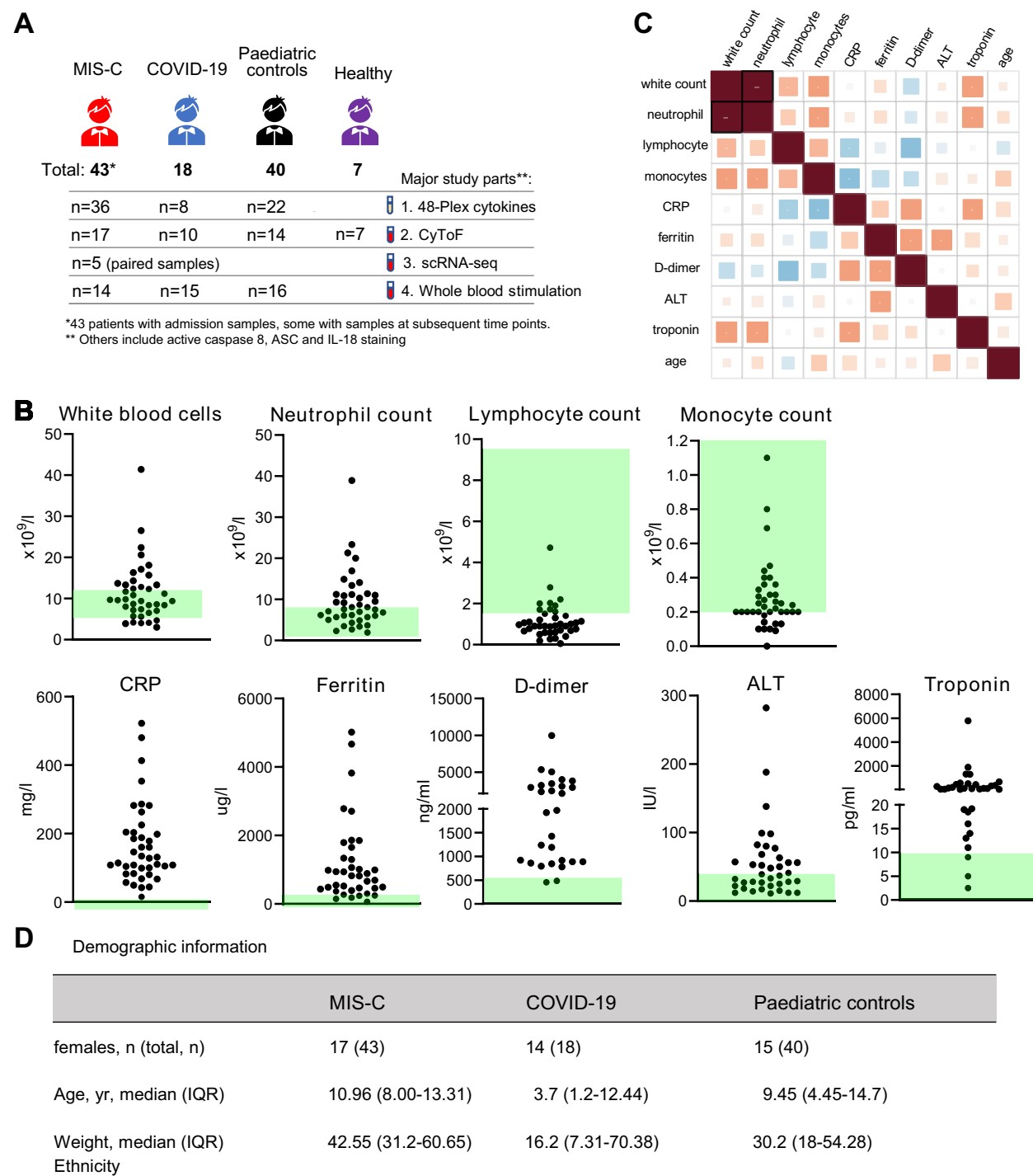

the widely reported TCR Vβ21.3 T cell expansion in MIS-C, we included an antibody specific for this TCR in the T cell panel. Based on the strong IFN-γ signature in MIS-C patients, in our study and in others, we also included an antibody specific for the IL-18 alpha receptor (IL-18R) since IL-18 is well known to stimulate T cell production of IFN-γ[26]. PBMCs from five paediatric groups were examined: (i) healthy

volunteers, (ii) acute MIS-C, (iii) MIS-C follow-up, (iv) severe COVID-19 pneumonia, and (v) other infections (including respiratory infection, sepsis and appendicitis). Data cleaning and unmixing are shown in Supplementary Fig. 2 and manual gating in Flowjo software in Supplementary Fig. 3. Unsupervised exploratory analysis suggested changes in cell populations, such as decreased CD16+ NK cells

**Fig. 1 | Clinical test results of children with MIS-C. A** Graphic summary of the study. Children with MIS-C or COVID-19 pneumonia were studied alongside control paediatric patients and healthy volunteers. There were 4 major parts to the study: 48-plex biomarker assay of plasma samples, high dimensional CyToF study of cell surface markers, scRNA-Seq, and whole blood functional assays. Child icon was from Microsoft PowerPoint 2020. **B** Clinical test results of MIS-C patients including white blood cell, neutrophil, lymphocyte and monocyte counts, C-Reactive Protein (CRP), ferritin, D-dimer, alanine transaminase (ALT) and troponin levels. Green area indicates the reference range[i]. **C** Correlation matrix of clinical test results. Statistical significance ($p < 0.05$) is indicated with star symbols (without adjustment for multiple comparisons by FDR method unless there was a black box outline), with red and blue colours indicating positive and negative correlation respectively. **D** Table of demographic and ethnicity information for the study subjects. Median and interquartile range (IQR) are provided for age and weight data. Source data are provided as a Source Data file.

(Supplementary Fig. 4A, B) as well as levels of cell surface markers like CD95 and IL-18R (Supplementary Fig. 4C, D), comparing MIS-C to the healthy control group.

To get a global view of the T cell activation status, we first examined non-naïve T cells (NOT CD45RA+ CD27+ cells in CD56− CD3+ T cells) by t-distributed stochastic neighbour embedding (tSNE) dimensionality reduction (Fig. 3A, Supplementary Fig. 5A). Although we deliberately excluded TCR Vβ21.3 as a classifying parameter in this unsupervised analysis, T cells positive for this receptor still formed two distinct populations in the CD4+ and CD8+ compartment respectively (Fig. 3A, 1st panel). Compared to all other non-naïve T cells they were enriched for multiple T cell activation and exhaustion markers. Activation markers CD38 and HLA-DR levels were high in the TCR Vβ21.3+ T cells, with the latter more prominent in CD8+, but not CD4+ T cells (Fig. 3A, 2nd and 3rd rows). In contrast, levels of IL-18R and the co-stimulation receptors CD28 and ICOS were more prominent in TCR Vβ21.3+ CD4+ T cells than CD8+ T cells (Fig. 3A, 4th, 5th and 6th rows). A small proportion of TCR Vβ21.3+ CD8 and CD4 T cells expressed immune checkpoint markers PD-1 and CD39 (Fig. 3A, 7th and 8th rows). For other checkpoint markers, Tim-3 and LAG-3 expression was confined to a small proportion of Vβ21.3+ T cells and few of them expressed CD57, a marker of terminally differentiated T cells (Supplementary Fig. 4A).

Although the non-naïve T cells in the non-SARS-CoV-2 infection group lacked the expansion of TCR Vβ21.3 T cells present in MIS-C patients, they still showed similar expression of phenotype markers (Fig. 3A), particularly in the four sepsis cases of this group (Supplementary Fig. 5B). Therefore, our data suggest that activated T cells in patients with MIS-C and sepsis are phenotypically similar, despite differences in TCR usage.

We observed increased frequencies of TCR Vβ21.3+ T cells in most children with MIS-C, but not in those with respiratory COVID-19 or other infections, using mass cytometry data by human-supervised manual gating. These cells were enriched in activated HLA-DR+ CD38+ CD4+, CD8+ and double negative (DN) populations (Supplementary Fig. 5C–E). Measuring expression levels of multiple markers, including CD38, HLA-DR, ICOS and IL-18R, in non-naive T cells gave similar results to the unsupervised analysis. Notably, in MIS-C patients TCR Vβ21.3+ T cells expressed significantly higher levels of these markers than TCR Vβ21.3− T cells for both the CD4+ and CD8+ subsets (Fig. 3B).

Next, we explored possible differences between TCR Vβ21.3+ and TCR Vβ21.3− T cells within the activated T cells (HLA-DR+ CD38+). In MIS-C group, IL-18R and CD28 were both higher in TCR Vβ21.3+ T cells, compared to TCR Vβ21.3− T cells for both the CD4+ and CD8+ subsets (Fig. 3C). PD-1 and ICOS levels were higher in the TCR Vβ21.3+ T cells, compared to TCR Vβ21.3− T cell cells, but only for the CD4+ subset. In contrast, there were no differences in these markers between TCR Vβ21.3+ T cells, compared to TCR Vβ21.3− T cell cells in the infection group.

### Expanded CD4+ TCR Vβ21.3+ cells in MIS-C patients consist of Th1, Th17, Treg, but not Th2 cells

Assessing differentiation state, TCR Vβ21.3+ T cells in MIS-C patients were predominantly central memory and effector memory cells (Fig. 3D and Supplementary Fig. 5F). The CD4+ subset of TCR Vβ21.3+ T cells was enriched for Th1, Th17 and Treg cells in MIS-C patients whereas the much smaller number of TCR Vβ21.3+ T cells cells in the control donors lacked this enrichment (Fig. 3E). This is consistent with the plasma cytokine profiles we observed in these patients (Fig. 2). For total CD4+ T cells, there was no difference in T helper subset distribution between any of the patient groups (Supplementary Fig. 5G–J, Th1, Th2, Th17 and Treg, respectively). There was no apparent difference in the activation status of other T cell populations such as double negative (DN), mucosal-associated invariant T (MAIT) and γδT cells in MIS-C compared to other groups (Supplementary Fig. 5K–M, respectively).

### CD95 and IL-18R levels are increased in CD16+ NK and monocytes in MIS-C

Total NK cell frequency and the ratio of CD16+ to CD16− NK cells were similar across groups using manual human-supervised analysis of innate immune cells (Fig. 4A, B). However, CD16+ NK cells in children with MIS-C had higher levels of IL-18R and CD95 consistent with an activated or inflammatory state (Fig. 4C, D).

Monocyte activation with changed surface markers has previously been reported in MIS-C[22,27]. Manual gating of monocytes in our mass cytometry data found no significant difference in the proportion of the three canonical monocyte subsets (classical, intermediate and non-classical[28]) in MIS-C patients, healthy donors or any of the other infections (Fig. 4E–G). Decreased levels of CD14, CD4 and HLA-DR were found in classical monocytes of MIS-C and SARS-CoV-2, non-SARS-CoV-2 infection conditions, compared to healthy volunteers (Fig. 4H–J). We observed other monocyte alterations that were shared between patients with MIS-C and SARS-CoV-2, non-SARS-CoV-2 infection conditions, including increased levels of CD63 and CD64, and decreased levels of CD11c, CD36, and CD86 (Fig. 4L–O). In contrast, increased CD95 levels on classical monocytes were specific to MIS-C (Fig. 4K).

We explored if cytometry data could be used to guide MIS-C diagnosis. IL-18R levels on non-naïve TCR Vβ21.3+ CD4+ T and CD16+ NK cells separated MIS-C ($n = 17$) from acute infection ($n = 14$) (AUC > 0.9 for all the markers, Fig. 4P), similar to CD38 and ICOS levels on non-naïve TCR Vβ21.3+ CD4+ T cells. These markers were better than plasma cytokine levels (such as CXCL10, IL-10) for differentiating MIS-C from infection controls (Supplementary Fig. 6A, B). It is of note that the discriminatory power of these markers was lower for the CD8+ subset of non-naïve TCR Vβ21.3+ T cells compared to the CD4+ subset (IL-18R, CD38 and ICOS in Supplementary Fig. 6C–E, respectively).

The increased levels of IL-18R we observed on the TCR Vβ21.3+ T cells and CD16+ NK cells of MIS-C patients suggested activation of this pathway[29]. In MIS-C patients, plasma IL-18 levels significantly correlated with the frequency of TCR Vβ21.3+ T cells in the total CD3+ T cell population, which was largely due to the CD4+ T cell subset (Fig. 4Q). We also observed that in MIS-C patients, IL-18R levels in all CD4+ T cells also significantly correlated with plasma IL-18 levels.

### Functional analysis of MIS-C blood found no evidence of overactive NLRP3 inflammasome activation but increased caspase 8 activity

The above data showed striking changes in the immune cells, in particular T cells in MIS-C, with similarities to TSS. However, there

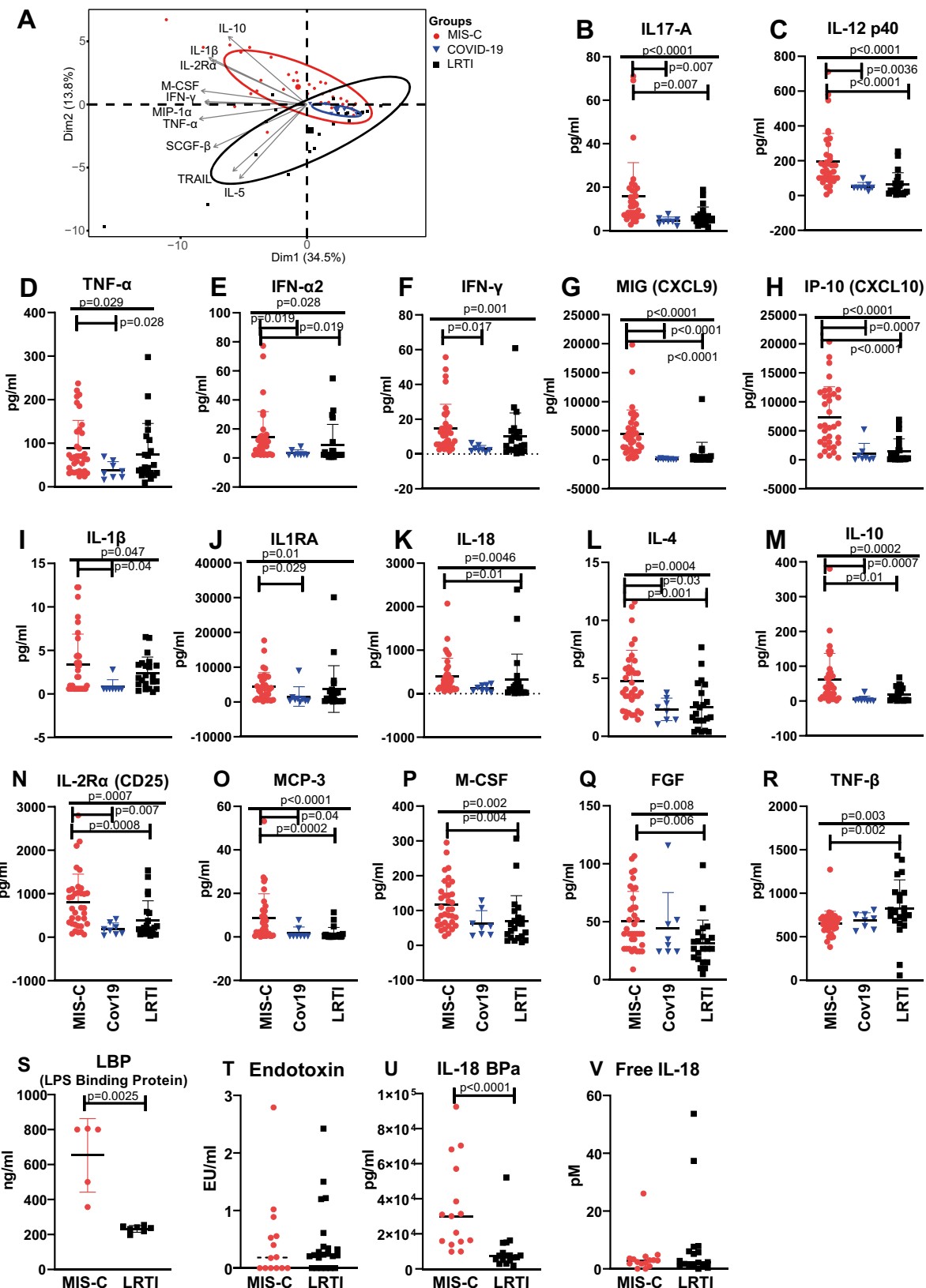

was no any clinical evidence of active infections in these children. Another explanation of the key MIS-C feature – long-lasting fever – is an autoimmune disease. A group of these diseases are associated with overactive NLRP3 inflammasome signalling and could be diagnosed with a whole blood stimulation assay[30]. Our data also suggested an important role for IL-18 signalling in MIS-C and it prompted

us to examine NLRP3 inflammasome activation, the classical IL-18 production pathway.

As expected, addition of LPS plus ATP induced high levels of IL-1β in the blood of non-MIS-C patients (paediatric admission and COVID-19 pneumonia) and children recovered from MIS-C at follow-up stage, at both the 4 h and 24 h stimulated whole blood time points (Fig. 5A). LPS

**Fig. 2 | Multiplex cytokine analysis identifies prominent Th1 plasma inflammatory markers in acute MIS-C.** Plasma cytokine and chemokine levels were measured for 3 groups of acutely ill paediatric patients: MIS-C, PICU COVID-19 pneumonia and LRTI patients ($n = 38$, $n = 8$ and $n = 22$ respectively). Horizontal line indicates the median value of each group. **A** PCA analysis of 48-plex Luminex assay results showing the ten most variable analytes; different shapes & colours are used for each patient group. **B–R** Results for 17 of the 48 cytokines and chemokines analysed, including IL-12 p40, IL-18 and IFN-γ. Star symbols above a straight line denoted significant changes for the initial Kruskal–Wallis test while those above the capped line denote significant changes for the subsequent Dunn's multiple comparison test. **S** LBP levels in acute plasma samples of MIS-C and LRTI groups ($n = 5$ and 7, respectively. $P = 0.0025$). **T** LPS levels in acute plasma samples of MIS-C and respiratory groups ($n = 14$ and 22, respectively. $P = 0.65$). **U** IL-18BPa levels in acute plasma samples of MIS-C and LRTI groups. ($n = 15$ and 17, respectively. $P < 0.0001$). **V** Free IL-18 levels calculated by comparing IL-18 and IL-18 BPa levels for the samples analysed in (U); $p = 0.77$. Data shown in (**S**–**V**) were analysed by Mann–Whitney test. Horizontal lines in the graphs indicate the median for each group. Source data are provided as a Source Data file.

plus ATP also increased levels of IL-18 in blood samples from the paediatric admission patients, although only at the 24 h post-stimulation time point (Fig. 5B). However, for MIS-C patients, LPS plus ATP induced a much smaller increase in IL-1β and IL-18 levels were unchanged. Of note, their IL-18 levels were high across all conditions including the unstimulated control (Fig. 5A, B). TNF-α induction by LPS alone in the whole blood of the MIS-C group was also much weaker than that observed in blood samples from the control respiratory COVID-19 patients and other paediatric admission patients (Fig. 5C).

Given that only 3 out of 14 MIS-C admission samples (showed in open circles of Fig. 5A–C) in this assay were from patients not pre-treated with immune-suppressing drugs such as glucocorticoids or intravenous immunoglobin (IVIG), treatment effects could have influenced our results. To address this issue, we used two other strategies. The first was to examine NLRP3 activation markers, instead of functional readout in MIS-C samples without prior glucocorticoid/IVIG treatments. The second one was to examine this pathway in MIS-C follow-up samples when there was no treatment effect anymore.

We performed immunostaining of ASC (apoptosis-associated speck-like protein containing a CARD) specks, a classic NLRP3 activation marker. In fixed MIS-C blood samples, we observed few ASC speck positive cells, with similar levels between MIS-C admission and discharge stages, much lower than the positive control samples treated with LPS plus ATP (Supplementary Fig. 6F).

Secondly, we undertook a whole blood assay on convalescent blood samples from a subset of children with MIS-C collected at around 1 month after hospital discharge. This analysis showed there was no difference in stimulation response between MIS-C follow-up samples and the control paediatric admission patient group (for IL-1β, IL-18 and TNF-α, Fig. 5A–C, respectively) and therefore no evidence of an inherent overactive NLRP3 pathway in MIS-C.

In the absence of evidence for NLRP3 overactivation in MIS-C, we investigated other signalling pathways to explain high IL-18 activity. It has previously been shown that high CD95 levels in monocytes can lead to NLRP3-independent, caspase 8-dependent, release of IL-18[31]. As we observed high CD95 levels on monocytes in MIS-C blood samples, we measured their downstream caspase 8 activity. Compared to COVID-19 monocytes, there was indeed significantly increased active caspase 8 levels in MIS-C samples (Fig. 5D).

### Increased *IL-18* mRNA levels in CD16- NK cells of MIS-C

To explore the above changes at the transcriptome level, we analysed paired PBMC samples from five children with MIS-C (collected at the acute and follow-up stages) by scRNA-seq. Major populations were identified by Uniform Manifold Approximation and Projection (UMAP, Fig. 6A). Similar to the CyToF data (supplementary Fig. 4), there were increased proportions of low-density neutrophils, immature B cells and decreased CD16+ NK, cytotoxic CD8+ T cells in acute MIS-C samples (Fig. 6B). The expanded TCR Vβ21.3+ T cells that we detected in MIS-C patients by CyToF, were also evident in the TCR sequences of the PBMCs collected during acute MIS-C (Supplementary Fig. 7A). TCR Vβ 21.3 was not paired with a specific TCR Vj chain in MIS-C (Supplementary Fig. 7B), indicating a superantigen effect. They had an

activated phenotype, with increased expression of genes such as *CD27*, *CD28* and *CTLA-4* (supplementary Fig. 7C). When MIS-C non-naïve CD4+ T cells were closely examined, TCR Vβ21.3+ T cells were enriched in cytotoxic, activated Th1 and cycling T cells (Fig. 6C), similar to findings from the CyToF experiment.

Regarding monocytes, we found decreased levels of M1 markers like CD86, HLA-DRB1, and increased levels of M2 markers (linked to anti-inflammatory state of macrophages) including *IL1R2*, *PPARG*, *CD163*, *MRC1* and *IL-10* in MIS-C acute samples (Fig. 6D). This is consistent with the reduced inflammatory response observed in the whole blood stimulation experiment.

To identify the cellular source of *IL-18*, we mapped its expression in UMAP, alongside *IL1B*. While *IL1B* expression was confined to monocytes, *IL-18* expression was more ambiguous. Nevertheless, it was clear that CD16- NK cells also express high *IL-18* in MIS-C, in addition to monocytes (Fig. 6E). Further analysis showed that CD16- NK cells had significantly higher levels of *IL-18* transcripts at acute compared to follow-up stage, but not monocytes (Fig. 6F). Immature status of the CD16- NK cells was evidenced with higher NCAM1 (encoding CD56) and KLRC1 expression and lower FCGR3A (encoding CD16) and CX3CR1 expression compared with mature CD16+ NK cells (Supplementary Fig. 7E). The significant difference remained after removing small numbers of contaminating hematopoietic stem cells (cluster 4 and 5 in Supplementary Fig. 7F, with stem cell markers coloured in 7G). We verified this observation at the protein level using flow cytometry. In MIS-C, CD56+ NK cells (the same cell population as the CD16− cells in the scRNA-seq analysis) displayed high IL-18 levels, similar to those of myeloid cells (Fig. 6G).

Finally, our scRNA-seq data enabled two further observations. First, we found that the MIS-C patient with the most severe disease (requiring 10 days of inotropes support), expressed HLA-A*02,B*35,C*04, the same alleles reported to be associated with both MIS-C severity and TRBV-11-2 expansion by others[3] (Supplementary Table 3). Second, to examine for possible immune abnormalities predisposing MIS-C development, we compared the data from PBMCs collected at the follow-up stage of MIS-C to data from healthy paediatric PBMC samples[32]. This analysis revealed a similar immune cell distribution for these two groups, although there were more monocytes with an interferon-stimulated signature and a lower frequency of CD16- NK cells in the MIS-C follow-up samples (Fig. 6G).

## Discussion

We combined high dimensional analysis of systemic cytokines and cellular immunity with functional immune assays to investigate immunity in children with MIS-C, comparing them to acute respiratory COVID-19, and acute non-SARS-CoV-2 related infections. Our data provide new insights into the expanded TCR Vβ21.3+ T cells that are a hallmark of MIS-C and suggest a key role for IL-18. Unexpectedly, in MIS-C patients, IL-18 does not appear to be produced by an overactive NLRP3 inflammasome pathway. Instead, we observed high CD95 levels in monocytes and high downstream caspase 8 activity, which has previously been shown to activate IL-18 production. Consistent with these observations, we detected IL-18 transcripts and protein in monocytes from acutely ill MIS-C patients. Surprisingly, we also found

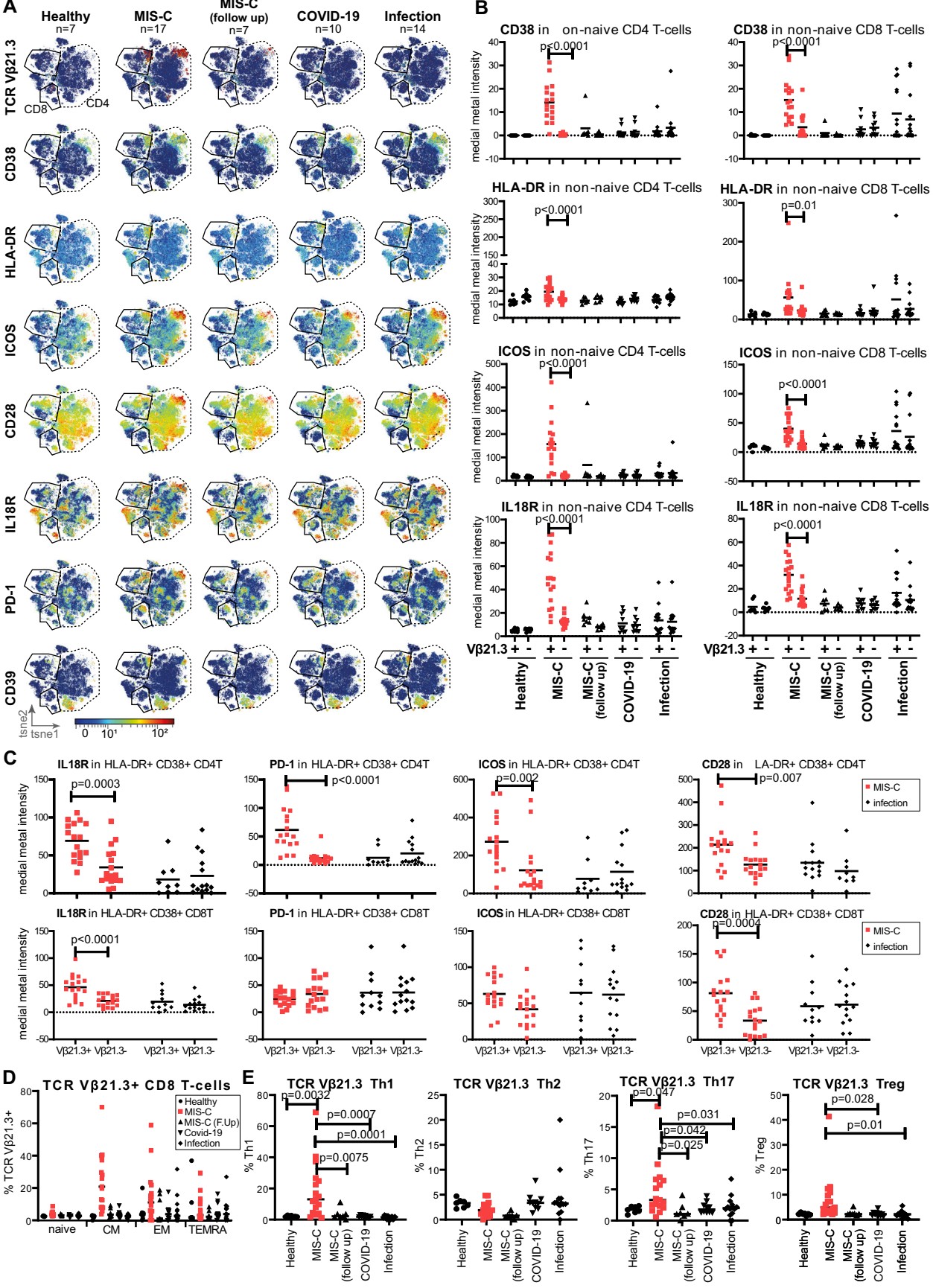

**Fig. 3 | Deep CyToF phenotyping of TCR Vβ21.3 T cells in MIS-C.** Cryopreserved PBMCs from healthy volunteers (n = 7) or patients with: acute MIS-C (n = 17), MIS-C follow-up (n = 7), PICU COVID-19 pneumonia (n = 10) or acute paediatric infection (n = 14, mixed chest, gastrointestinal and systemic infections) were analysed by CyTOF. **A** Opt-SNE plotting of non-naïve CD3+ T cells (selected by Boolean gating of CD27+ CD45RA+ cells) using a maximum of 10,000 cells per sample. Note that the TCR Vβ21.3 parameter was not used to construct the tSNE projection. For each plot, colours show expression levels of the following markers from each group: TCR Vβ21.3, CD38, HLA-DR, ICOS, CD28, IL-18R, PD-1 and CD39. **B** Median metal staining intensities of CD38, HLA-DR, ICOS and IL-18R were quantified in TCR Vβ21.3± non-naïve CD4 and CD8 T cells identified by human gating of the data. Two-way ANOVA was conducted with Šídák's multiple comparisons test to compare between TCR Vβ21.3+ and TCR Vβ21.3– cells for each group. **C** Comparison of IL-18R, PD-1, ICOS and CD28 levels between TCR Vβ21.3+ and TCR Vβ21.3 activated T cells (cells positive for HLA-DR+ and CD38+) Two-way ANOVA was conducted with Šídák's multiple comparisons test to compare between TCR Vβ21.3+ and – cells for each group. **D** Percentage of TCR Vβ21.3+ cells in CD8 T cells (divided by differentiation state, based on CD45RA and CD27 levels) subsets. Ordinary one-way ANOVA test was used to compare the five groups, with Dunn's multiple comparisons test to compare each group to the MIS-C group. Naïve T cells were CD45RA+ CD27+, central memory (CM) CD45RA– CD27+, effector memory (EM) CD45RA– CD27– and terminal effector (TE) CD45RA+ CD27. MIS-C (F.Up): MIS-C follow-up. **E** Percentage of TCR Vβ21.3+ cells in CD4+ T cells (divided into Th1, Th2, Th17 and Treg subsets based on chemokine receptor expression). Ordinary one-way ANOVA test was used to compare the five groups, with Dunn's multiple comparisons test to compare each group to the MIS-C group. Horizontal lines in graphs in (**B**–**E**) indicate the median for each group. Source data are provided as a Source Data file.

high *IL-18* transcripts and IL-18 protein in the CD16- NK cells of these patients.

Our cytokine array data confirmed previous findings in MIS-C patients of increased plasma IL-1 family cytokines (including IL-1 and IL-18), IFN-γ signatures (such as downstream CXCL9, CXCL10) as well as the anti-inflammatory cytokine IL-10, IL-18 BP[2]. Gut epithelial damage was explored as a mechanism of inflammation in MIS-C[33]. Consistent with the study, we detected increased plasma levels of LBP in our MIS-C patient cohort. LBP is secreted by the liver in response to acute stress, like increased endotoxin in the blood. However, our MIS-C patients did not have significantly raised levels of plasma endotoxin compared to PICU LRTI patients, like in a previous study[2]. An alternative explanation for the increased IFN-γ signature could be increased levels of IL-18[17], which is raised in MIS-C patients and is a classical inducer of IFN-γ secretion[26]. Our MIS-C patients similarly had high levels of plasma IL-18 levels. However, they also had high levels of IL-18-BP; when this inhibitor of IL-18 was taken into account, levels of biologically active free IL-18 in MIS-C patients were equivalent to those of children with non-SARS-CoV-2 respiratory infections and lower than those reported previously for children with respiratory COVID-19[34]. Whether persistent IL-18 response after COVID-19 infection is a trigger for MIS-C awaits future examination.

Our detailed phenotyping of TCR Vβ21.3 T cells, which are present at high frequency in MIS-C patients, showed they occupied multiple T cell compartments, including the CD4+, CD8+ and double negative subsets. In terms of activation state, TCR Vβ21.3 T cells in MIS-C patients were heavily skewed towards the central memory and effector memory states. However, because most T cells in children are naïve, numerically most Vβ21.3+ T cells in the MIS-C patients were naïve. This suggests that superantigen stimulation alone, without co-stimulating receptor engagement, is insufficient to fully activate the TCR Vβ21.3+ T cell compartment in MIS-C.

T cells in MIS-C patients have been reported to express exhaustion markers such as PD-1, CD39 and LAG-3[5,20], which were mainly focused in CD8+ T cells. Our data now shows that in MIS-C patients, the majority of T cells with this phenotype are comprised of expanded non-naïve TCR Vβ21.3+ T cells, in both CD4+ and CD8+ T cells. Previous studies in MIS-C focused on using HLA-DR+ CD38+ as markers as T cell activation, leading to oversight of the TCR Vβ21.3+ CD4+ T cells, which had weaker HLA-DR expression. In our study, we found TCR Vβ21.3+ CD4+ T are strongly changed in MIS-C, with high levels of co-stimulation receptors like ICOS, CD28 and IL-18R. CD28 is known to play an important role in bacterial superantigen-stimulated T cell activation[35,36] and this may also be the case for MIS-C. IL-18R is also a key contributor to superantigen-stimulated T cell activation, based on a gene knockout study in mice[37]. These receptors were not studied before to the best of our knowledge. They serve as good markers to differentiate MIS-C with other infections in TCR Vβ21.3+ CD4 T, compared to TCR Vβ21.3+ CD8 T cells. In CD4+ T cells, we found expanded Vβ21.3+ T cells in the

Th1, Th17 and Treg compartments. This is the first time that increased TCR Vβ21.3+ Treg cells were reported in MIS-C. It could explain the transient immune activation nature in MIS-C[20].

We note that TCR Vβ21.3+ T cell phenotype resembles that seen in other superantigen-related conditions such as TSS[38]. However, our data showed MAIT cells were only modestly activated in MIS-C, which contrasts to data from bacterial superantigen models[39], and could reflect the MIS-C superantigen being viral rather than bacterial in origin.

Our functional assays revealed that canonical NLRP3 activation is not overactive in MIS-C at the acute or follow-up stage. This differs from the proposed NLRP3 pathway activation seen in Kawasaki disease where transcriptional changes in the genes associated with the NLRP3 pathway were identified[40]. Increased monocyte *IL1B* levels in a Kawasaki disease scRNA-Seq study[41] is also not found in MIS-C here. Therefore, our data suggest the NLRP3 pathway is unlikely to increase the risk of vascular damage in MIS-C patients in the long term and may explain the lack of long-term complications seen 6 months after MIS-C diagnosis[42]. It is consistent with the much lower levels of circulating endothelial cells present in MIS-C patients compared to those with Kawasaki disease[43]. However, we cannot exclude a role for this pathway earlier in the course of disease before MIS-C symptoms develop, since NLRP3 activation has been reported in symptomatic adult COVID-19 patients[44]. Our findings contrast with a previous study which compared inflammatory profiles of children with Kawasaki disease to samples from three children with MIS-C. In these MIS-C patients, increased inflammasome activity was found when measuring caspase 4 and caspase 1 cleavage in granulocytes[45]. Activation of caspase 1 could be downstream of inflammasome activating receptors such as AIM2, PYRIN, NLRC4 and NLRP1, not just NLRP3[46]. IL-1β cytokine levels, another readout of inflammasome activity[47], have not been demonstrated to be high in children with MIS-C[2,11,17,45], although this may be confounded by the treatment effect. The largest randomised trial of immunomodulatory treatments in MIS-C (PIMS-TS, MIS-C; RECOVERY) found neither IVIG nor anakinra (IL-1 receptor antagonist) had any effect on the duration of hospital stay compared with usual care[48]. Examining thus whether inflammasomes are over-activated in these patients requires further work in a larger number of patients.

NLRP3 activation in myeloid cells is a major source of IL-18. The lack of NLRP3 activity we detected in MIS-C patients raises the question of the source of their high IL-18 levels. Alternative sources include NLRP3-independent IL-18 production[49] or extracellular IL-18 activation by various proteinases[50]. For example, CD95-activated caspase 8 has previously been shown to induce IL-18 release independently of NLRP3[31]. This mechanism could operate in MIS-C, as we detected CD95 upregulation in both the monocytes and NK cells of acute MIS-C patients, as well as the downstream active caspase 8, a marker of apoptosis. Another source of IL-18 in MIS-C patients could be CD16- NK cells, which we found express high levels of *IL-18* mRNA during the acute phase of illness, then decrease upon recovery. We validated

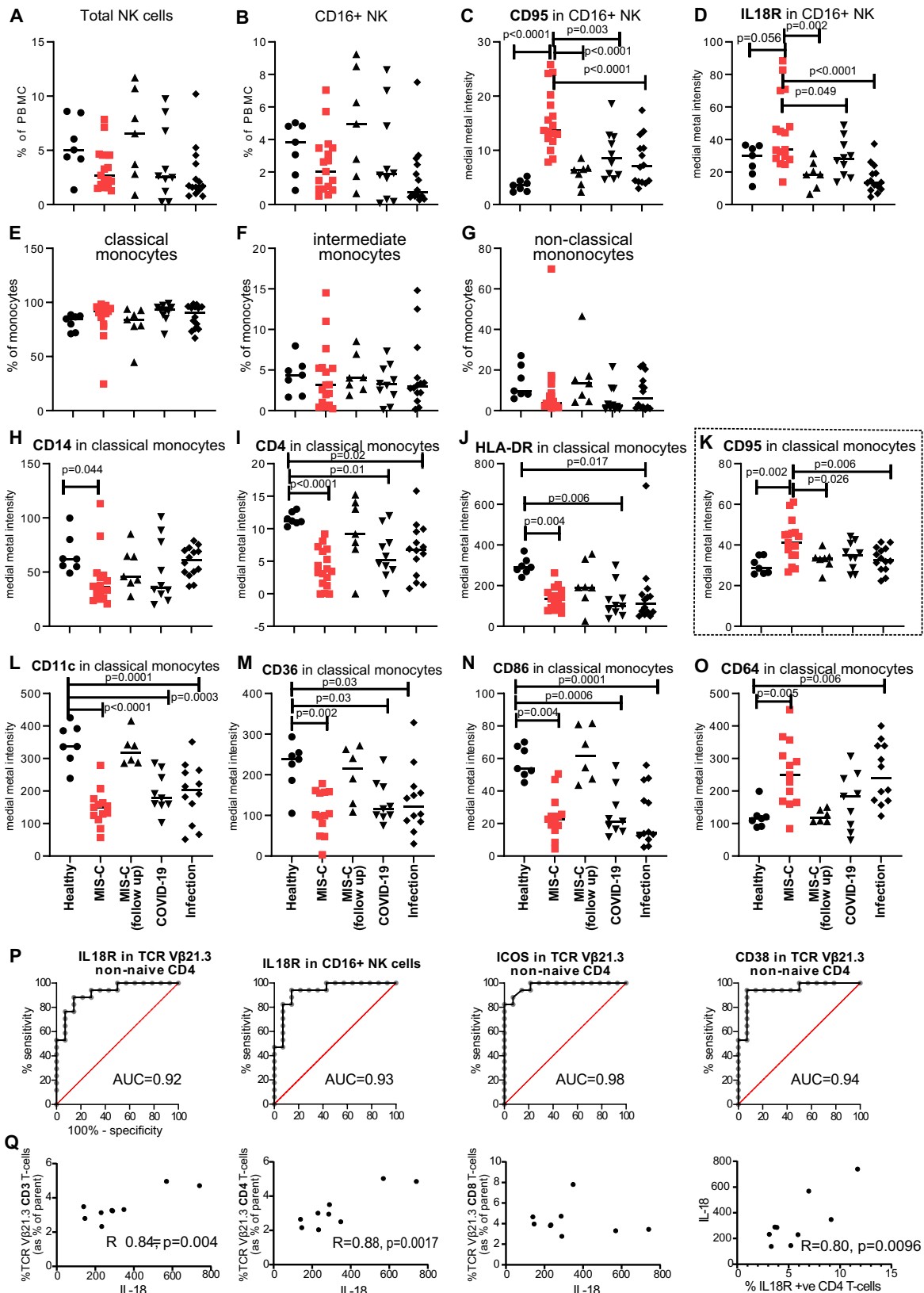

these IL-18 transcripts were expressed as protein. Although these cells can produce several cytokines[51,52], to our knowledge IL-18 production has not been reported. Of note, the levels of total IL-18 in the plasma of MIS-C patients are much lower than in other IL-18-mediated diseases such as macrophage activation syndrome[53]. and hemophagocytic lymphohistiocytosis[23]. Whether this reflects different Il-18 activation

pathways (caspase 8 mediated in MIS-C, NLRP3 mediated in others), or a different cellular source of IL-18 (CD16- NK versus monocytes) is currently unknown.

Our study has some limitations. Due to the small sample volumes that can be collected from sick children, we used whole blood samples in the functional assay. Although this means the results could be

**Fig. 4 | CD16+ NK cells and monocytes of acute MIS-C patients have increased cell surface levels of IL-18R and CD95.** Monocyte and NK cell populations were analysed in (**A**–**K**) in the same experiment groups shown in Fig. 3: acute MIS-C ($n = 17$), MIS-C follow-up ($n = 7$), PICU COVID-19 pneumonia ($n = 10$) or acute paediatric infection ($n = 14$, mixed chest, gastrointestinal and systemic infections). Frequency in PBMCs of **A** total NK cells or **B** CD16+ NK cells and median metal staining intensity of **C** CD95 or **D** IL-18R on CD16+ NK cells. **E**–**G** show the proportion of each subject's monocytes classified into the three canonical subsets: classical (CD14++ CD16−), intermediate (CD14+ CD16+), or non-classical (CD14+ CD16++) monocytes. **H**–**O** show the MFI of the indicated proteins on the surface of classical monocytes using data from the T cell (**H**–**K**) or monocyte (**L**–**O**) CyTOF antibody panels (healthy children ($n = 7$), acute MIS-C ($n = 13$), MIS-C follow-up ($n = 6$), acute paediatric COVID-19 pneumonia ($n = 9$) or acute paediatric infection

patients ($n = 12$, mixed chest, gastrointestinal and systemic infections)). Horizontal line indicates the median value of each group tested. Statistical testing was performed using ordinary one-way ANOVA; results of Dunnett's multiple comparisons test comparing the acute MIS-C group to all other groups is shown in (**A**–**D**, **K**) and comparing healthy children to all other groups in (**H**–**J**, **L**–**O**). **P** Receiver operating characteristic curves of the indicated markers on particular cell subsets for MIS-C diagnosis ($n = 17$, acute samples) with infection samples as control ($n = 14$). Area under the curve values are shown in each graph. All four marker/cell combinations were significantly different between the two groups. **Q** Correlation between plasma IL-18 levels and frequency of the indicated cell populations. Pearson correlation coefficient and significance are shown on each graph. $N = 9$. Source data are provided as a Source Data file.

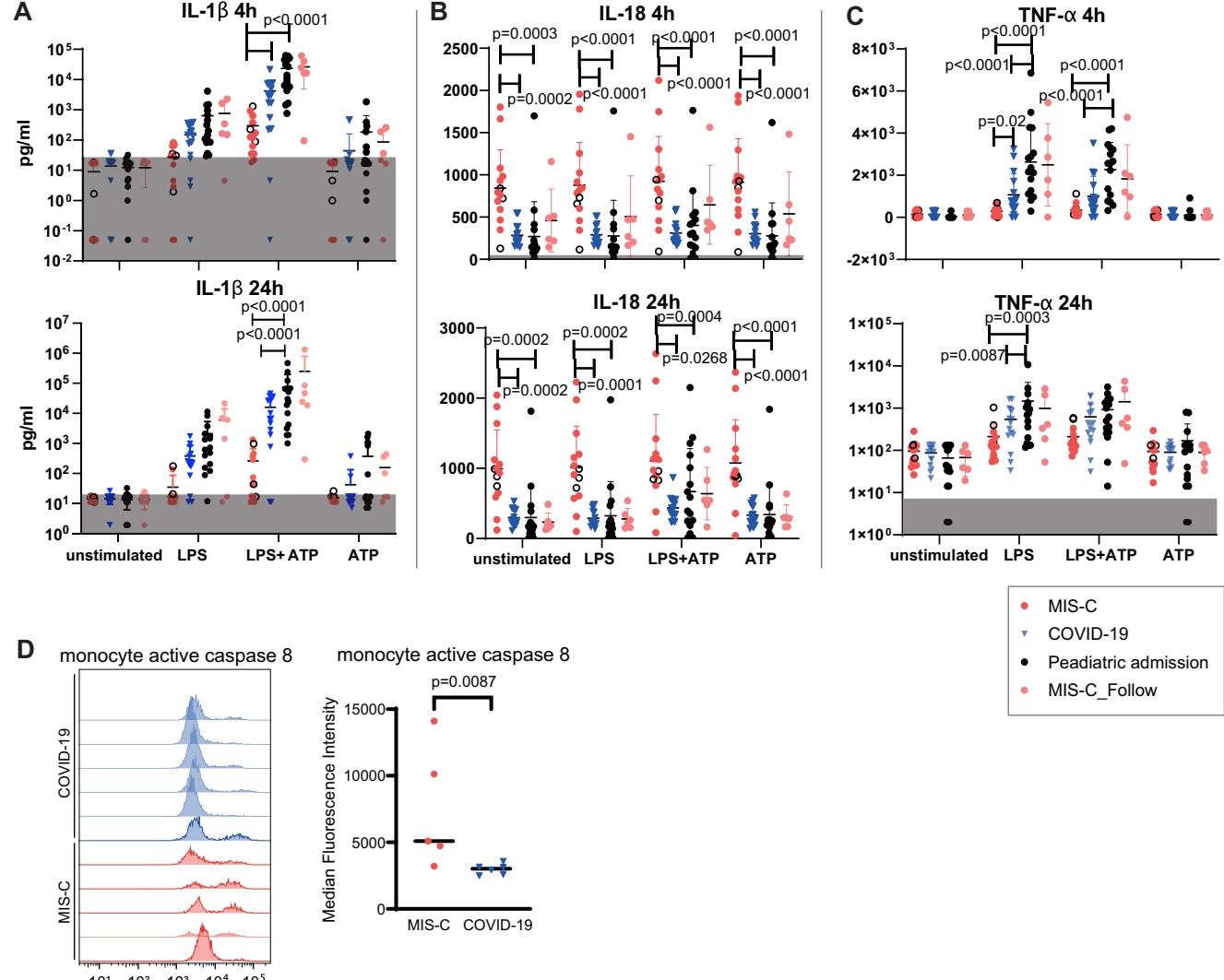

**Fig. 5 | Lack of overaction of NLRP3 inflammation activation in MIS-C and increased active caspase 8 activity in MIS-C monocytes.** Whole blood samples from children with MIS-C ($n = 14$), acute COVID-19 pneumonia (COVID-19, $n = 15$), other paediatric admissions (Paediatric Admission, $n = 16$) and MIS-C follow-up patients (MIS-C_Follow, collected about 1 month post hospital discharge, $n = 6$) were stimulated in vitro with LPS (50 ng/ml), ATP (5 mM), or LPS plus ATP (50 ng/ml and 5 mM respectively, ATP was added 3 h after the LPS addition). Cytokines were measured 4 h and 24 h after stimulation. Levels of IL-1β, IL-18 and TNF-α levels were shown in (**A**–**C**). The lower limit of quantification of IL-1β is about 18 pg/ml, 16.5 pg/ml for IL-18 and 7.3 pg/ml for TNF-α (grey-shaded areas). In the MIS-C group, samples with glucocorticoids/IVIG treatment in the last 24 h before sampling were

marked as unfilled circle. Horizontal line and error bars show the mean and standard deviation. Results of two-way ANOVA with post hoc Tukey's multiple comparisons test between the first 3 groups (all collected at the acute stage) within each treatment condition are shown. There were no significant differences between the MIS-C-Follow and paediatric admission group by multiple Mann–Whitney tests with each treatment condition. **D** Monocyte active caspase staining signals from 5 MIS-C and 6 COVID-19 blood monocyte samples. Left: histograms of active caspase 8 staining (red lines indicate MIS-C, blue COVID-19). Right: Quantification of active caspase 8 median fluorescence intensity. Mann–Whitney test was used to compare the 2 groups. Source data are provided as a Source Data file.

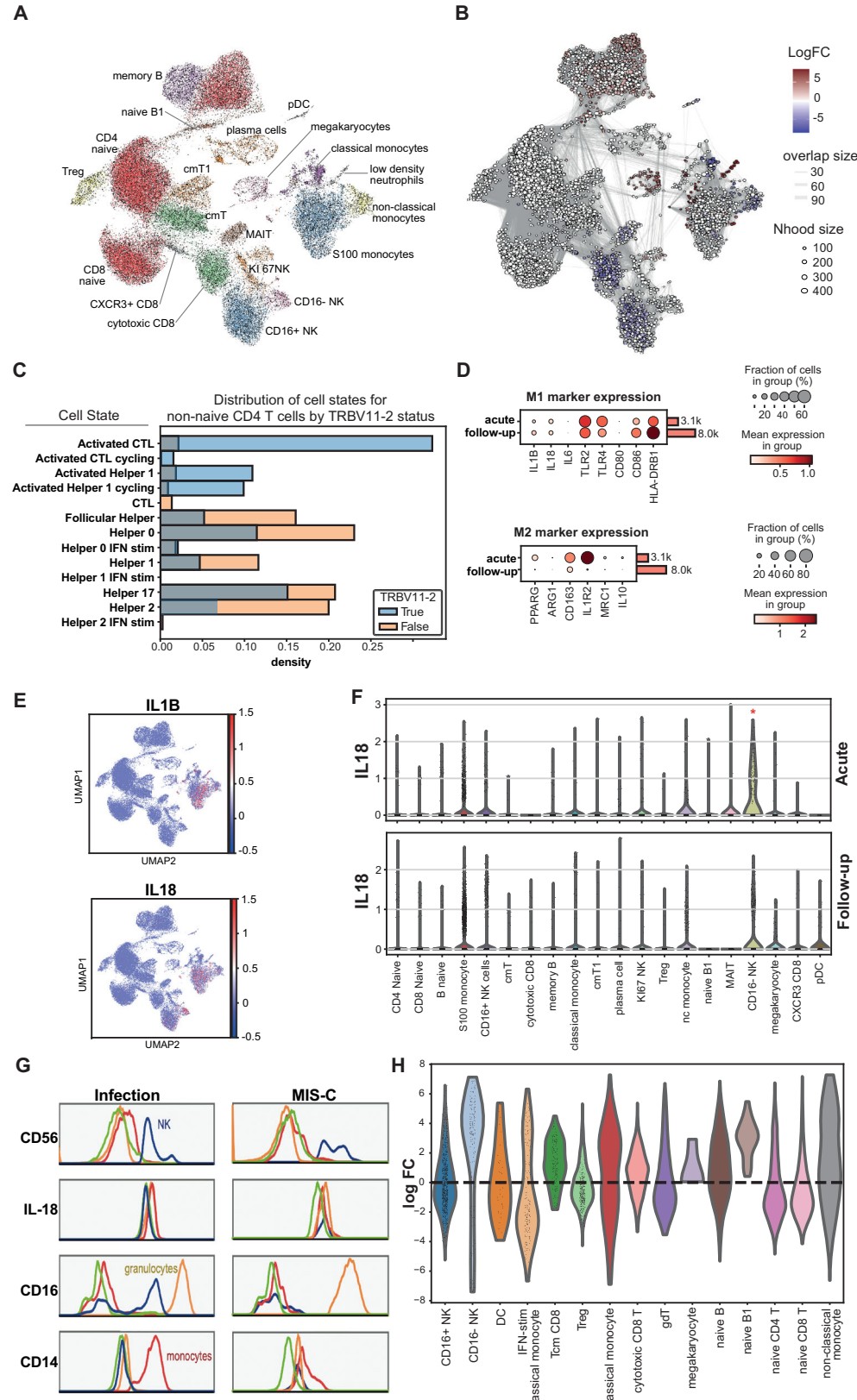

confounded by the effects of circulating cytokines or glucocorticoids, they provide a more accurate insight into the in vivo status of the cells and the responsiveness of the relevant signalling pathways. In addition, limited amounts of PBMCs restricted more in vitro studies, like caspase 8 blocking. Our cohort of MIS-C patients was recruited from intensive care or high-dependency units. This means a proportion of

our cohort had received glucocorticoid/intravenous immunoglobulin treatment prior to samples being taken. Future studies with samples from untreated patients would avoid the confounding effects of previous immune-modulating therapies.

In summary, our data suggest IL-18 signalling could link activated innate and adaptive immune cells in MIS-C, facilitating

**Fig. 6 | scRNA-Seq reveals CD16- NK cells express high *IL-18* transcripts in acute MIS-C patients.** ScRNA-seq analysis of paired PBMC samples from five children with acute MIS-C and at follow-up (approximately one month later). **A** PBMC UMAP projection showing annotated cell populations **B** Differential cell abundance test result from miloR analysis, comparing acute and follow-up groups. Significantly changed clusters were colour indicated. **C** Distribution of TCR Vβ21.3+ and − T cells in different CD4 T subtypes. **D** selected M1, M2 marker expression in acute and follow-up monocyte populations (including the 4 different monocyte subtypes). **E** Expression of *IL1B* and *IL-18* mRNA transcripts in the UMAP. **F** Violin plots of IL-18 mRNA levels in different PBMC populations from MIS-C patients at the acute or follow-up stage. There was a significant change in IL-18 mRNA levels for the CD16- NK population when comparing admission and follow-up samples by Wilcoxon's tests (*q* value = 3.81E-06). **G** Intracellular staining of IL-18 in fixed blood samples. Left: staining in an infectious disease sample; right: example in an MIS-C (acute) sample. **H** Cell abundance comparing PBMC samples from healthy and MIS-C follow-up children. The healthy children PBMC data are from published dataset[2]. FC fold change, nc non-classical, cm central memory.

superantigen-mediated stimulation of T cells and activation of CD16+ NK cells, which seems independent of NLRP3 inflammasome activation. Either CD95 and caspase 8 mediated IL-18 production in monocytes or increased IL-18 production in CD16− NK could be the main source of IL-18 in MIS-C.

## Methods

### Study participants
The study enrolled children consented to either of the following ethically approved studies: the TrICICL study (reviewed and approved by South of Birmingham Research Ethics Committee, REC reference 17/WM/0453), or the DIAMONDS-Search study (reviewed and approved by London-Dulwich Research Ethics Committee, REC reference 20/HRA/1714) or the RASCALS study (reviewed and approved by Yorkshire and the Humber-Bradford Leeds Research Ethics Committee, REC reference 20/YH/0089). Informed consent was obtained (from parents or guardians of children <16 years of age) by trained health professionals.

Enrolment criteria included age up to 16 years and a clinical diagnosis of COVID-19-related illness requiring admission to PICU or high-dependency care units in England (Addenbrookes Hospital in Cambridge, Birmingham Children's Hospital in Birmingham and Great Ormond Street Hospital for Children in London). Control samples were obtained from children with febrile illness and severe respiratory inflammation conditions (non-COVID-19 related). Sex and gender information was recorded by self-reported information. And we didn't have consent for reporting disaggregated data for sex and gender at the individual level.

Enrolment was undertaken between May 2020 and January 2022. The UK Royal College of Paediatrics and Child Health's definition of MIS-C was used for diagnosis (https://www.rcpch.ac.uk/resources/paediatric-multisystem-inflammatory-syndrome-temporally-associated-covid-19-pims-guidance). COVID-19 paediatric patients in the study had positive nasal/oropharyngeal swab SARS-CoV-2 PCR test on admission and required admission into PICU due to COVID-19 pulmonary infection. Admission samples were collected within 24–48 h of hospital admission. For MIS-C patients, recovery samples were collected 48–96 h post-treatment, discharge samples collected upon discharge and follow-up samples about 1 month post-discharge. Note that some MIS-C patients were reported in a previous study from us[17].

### Blood sample processing
Peripheral blood samples were collected into EDTA (for Luminex assay and flow cytometry) or heparin tubes (for whole blood stimulation and PBMC harvest). Plasma was collected by centrifuging the EDTA anticoagulation blood for 10 min at 1000 × *g* at room temperature. Serum samples were processed by centrifuging the serum Vacutainer tube for 10 min at 1300 × *g* at room temperature. They were aliquoted and stored in the -80°C freezer. For cytodelics samples, 100 μl EDTA anticoagulation blood was mixed with 100 μl stabiliser solution (Whole blood processing kit / Gen 2, Cytodelics, Stockholm, Sweden). The tubes were shaken several times and left for 10 min at room temperature before being frozen at −80 °C. To collect PBMCs, whole blood was mixed with DPBS at a 1:1 ratio before being laid on top of a Leucosep tube containing 3 ml of Ficoll Paque Premium (GE17-5442-02,

Merck Life Science UK Limited, Dorset, UK). The tube was centrifuged for 10 min with brake off at 1000 × *g* at room temperature. After that, the top layer of a clear solution containing diluted plasma was stored in the −80 °C freezer. The PBMC pellet was washed twice before being resuspended in FBS solution containing 10% DMSO, with a final concentration ~1 × 10^6 cells/ml. Blood samples were usually processed within two hours of collection.

### Luminex assays and LBP, IL-18 BPa ELISA
The Bio-Plex Pro Human Cytokine Screening Panel (48-Plex #12007283, Bio-Rad Laboratories Ltd., Watford, UK) was used for the plasma and serum samples according to the manufacturer's protocol. In brief, samples were diluted 1 in 4 for the assay and standard controls were serially diluted 1 in 4 for 8 dilutions. A blank reference was used to check the background signal, an internal control to check the kit performance and a reference control for different batches. The plate was read on a Bio-plex 200 machine and the concentrations were calculated by the Bio-Plex Manager software. For samples falling out of the lower limit of the standard curve, the concentration was designated as the lowest extrapolated value.

A custom 4plex panel containing IL-1α, IL-1β, IL-18, IL-10 and a 3plex panel containing IL-6, TNF-α, IL-8 were used for the in vitro stimulation study (LXSAHM-04 and LXSAHM-03, R&D systems, Abingdon, UK). The supernatant was diluted 1 in 3 for the 4plex assay and 1 in 30 for the 3plex assay. The plate was read on a Bio-plex 200 machine and the concentrations were calculated by the Bio-Plex Manager software or on a Magpix machine with the xPONENT software. For samples falling out of the lower limit of the standard curve, the concentration was designated as the lowest extrapolated value.

Plasma LBP levels were measured with Human LBP DuoSet ELISA (DY870-05, R&D systems, Abingdon, UK) and IL-18BPa levels with Human IL-18 BPa Quantikine ELISA Kit (DBP180, R&D systems, Abingdon, UK), according to the enclosed protocols. Free IL-18 levels were calculated according to a published method[54].

### Plasma endotoxin detection
Plasma LPS was quantified using the ENDOLISA kit (SKU Number: 609033, BioMerieux). Plasma was transferred into LPS-free glass tubes (Lonza) and diluted in LPS-free water containing 1 mM LPS-free MgCl2 (Lonza). Standards were prepared as directed by the kit in water containing 1 mM LPS-free MgCl2. Plasma LPS was allowed to bind to the plate overnight with shaking at 37 °C. Plates were washed as directed. Fluorescence production was quantified using a CLARIOStar Plus spectrophotometer (BMG) with an excitation of 380 nm ± 20 nm and an emission of 440 ± 40 nm.

### CyToF experiment
Frozen PBMCs were thawed at 37° on the day of staining. The samples were washed once with DMEM 10% FBS and then counted using a Countess II machine (ThermoFisher Scientific, Oxford, UK). Generally, the cell viability was >80%. The cells were then washed again with FACS buffer (1% FBS in DPBS). Typical cell numbers were 0.5–2 million cells/sample. Samples were barcoded with CD45 antibodies labelled with one of three different metal isotypes without or with a

single isotope-labelled mixture of CD45 and B2M-specific antibodies. Six samples were pooled together into one tube after barcode labelling. Each tube contained an internal control PBMC sample for standardisation. In tubes 1–10 there were about 5.3 million cells based on original counting and 2.8 million cells in tubes 11–12. For tubes 1–10, the samples were split evenly into two tubes for T cell panel and monocyte panel staining, respectively. Only T cell panel staining was carried out for tubes 11–12. Cells were treated with TrueStain FcX blocking reagent (Biolegend) for ten minutes before antibody staining. Home-made antibody cocktails were filtered before use. For the T cell panel, TCR Vbeta21.3 was stained using a sequential three-step method (biotin-TCR Vbeta21.3 antibody, streptavidin-APC labelled with 170Er, anti-APC-170Er) to enhance its detection. The phenotyping antibodies were included in the third labelling step. Rhodium live/dead stain (Standard BioTools Inc, CA, USA) was used for subsequent dead cell exclusion. All staining was performed for 30 min at 4 °C. After incubation, the cells were washed twice and fixed with 1.6% formaldehyde in PBS overnight at 4 °C. Next day, cells were washed with PBS solution then incubated with 1:3000 diluted iridium in Maxpar fix perm buffer Solution (Product Number 201067, Standard BioTools Inc., CA, USA) at 4 °C. After 48 h, the samples were transferred into cyotubes and stored at −80 °C.

Samples were acquired using a Helios mass cytometer within 15 days of staining. On the day of acquisition, thawed cells were pelleted by centrifugation and washed with FACS buffer. Immediately prior to acquisition, cells were transferred into Maxpar Cell Acquisition Solution (Product Number 201240, Standard BioTools Inc., CA, USA) spiked with EQ calibration beads and filtered through a 70 µm cell strainer. Cell density in the solution was adjusted to collect about one million events per hour. The data were normalised using the EQ beads and exported as FCS3.0 files.

## CyToF data analysis
In Flowjo software (version 10.8.1, Flowjo, Oregon, USA), EQ beads were excluded and single live cells were gated. Then samples were unmixed by examining the barcode antibodies and fcs files were created for each sample. Data quality was inspected and initially analysed using CATALYST package[55] (release 3.16) in R (version 4.2.1). Unbiased analysis was carried out in OMIQ using opt-TSNE clustering with subsets of non-naïve T cells (maximum 10,000 non-CD27+ CD45RA+ CD3 T). Manual gating and analysis were performed using Flowjo software, and CD4 T cell subtype identification was conducted by examining cell surface chemokine receptors as reported before[56].

## LPS challenge and NLRP3 inflammation assay
The assays were based on a published protocol with slight modifications[30]. Whole blood samples were aliquoted onto 96 well plates, (140 µl/well, 8 wells in total). They were assigned to 4 groups in duplicates: (1) untreated, (2) LPS treatment (Ultrapure LPS, E. coli 0111:B4, InvivoGen, Toulouse France; 50 ng/ml final concentration), (3) LPS (50 ng/ml) plus ATP treatment (A6419, Merck Life Science UK Limited, Dorset, UK; 5 mM) and (4) ATP treatment (5 mM). DMEM solution (60 µl, 40 µl, 20 µl and 40 µl) was added into each well of the 4 groups respectively. The chemicals were made as 10× concentrated solution in 20 µl volume before being added in, with ATP added 3 h post LPS stimulation. Supernatants were collected at 4 h and 24 h, frozen at −20 °C before use in Luminex assays. Cell pellets were stored in 100 µl Trizol solution at −20 °C.

## ASC speck staining
Peripheral blood samples were collected into EDTA and aliquots (100 µl/tube) were mixed with Cytodelics storage solution (Whole blood processing kit / Gen 2, Cytodelics, Stockholm, Sweden) according to the manufacturer's protocol and stored in −80° freezers. Cell recovery was performed according to the protocol, with 15 min

fixation, 10 min of red blood cell lysis and then cell washing. Then cells were centrifuged onto glass slides by cytospin (300 g for 3 min). For the staining, cells were permeabilised with PBS-T (triton-X100, 0.1%), blocked with 10% FBS, and stained with ASC antibody for overnight in the cold room (1 in 100 dilution, anti-Asc, pAb (AL177), AG-25B-0006-C100, Adipogen AG, Fuellinsdorf, Switzerland). After that, the slides were washed and stained with secondary antibody (1 in 1000 dilution, Donkey anti-Rabbit IgG (H+L) Highly Cross-Adsorbed Secondary Antibody, Alexa Fluor 488, A-21206, Invitrogen, Renfrew, UK) for 1 h at cold room. The slides were stained with DAPI before coverslip mounting and imaging using a Leica SP5 microscope.

## Caspase 8 activity assay
Stored PBMC samples were thawed and washed in RPMI 1640 buffer with 10% FBS. After resuspending in RPMI 1640 media containing 10% FBS, the samples (290 µl/sample containing about $0.5 \times 10^6$ cells) were seeded into a well of a 24-well ultra-low attachment plate (Corning Costar Ultra-Low Attachment Multiple Well Plate, CLS3473, Merck Life Science UK Limited, Dorset, UK), together with 10 µl 30× caspase 8 assay buffer (FAM FLICA Caspase 8 Kit, ICT099, Bio-Rad Laboratories Ltd., Hertfordshire, UK). After 1 h in a 37 °C 5% CO2 incubator, the cells were analysed by flow cytometry. In brief, the cells were washed with 1× apoptosis washing buffer (from the same FAM FLICA Caspase 8 Kit as above) before being stained with anti-human CD14-Alexa 594 antibody (325630, BioLegend, London, UK) for 15 min in dark on ice. After washing twice, the samples were acquired on a BD Fortessa cytometer. Monocytes were gated as CD14 positive, side scatter high. Negative control cells were stained with only CD14 without caspase 8 assay buffer. Positive control cells were created by treating with 100 µM nigericin together with other reagents in the caspase 8 assay kit.

## Intracellular IL-18 flow cytometry assay
Peripheral blood samples were collected into EDTA and aliquots (100 µl/tube) were mixed with Cytodelics storage solution (Whole blood processing kit / Gen 2, Cytodelics, Stockholm, Sweden) according to the manufacturer's protocol and stored in −80 °C freezers. Cell recovery was performed according to the manufacturer's protocol, with 15 min fixation, 10 min red blood cell lysis and then washed before staining. To stain IL-18, cells were first permeabilized with 1× wash perm buffer (Intracellular Staining Permeabilization Wash Buffer (10X), 421002, Biolegend, London, UK) according to the manufacturer's protocol. Briefly, cells were washed twice in the perm wash buffer and resuspended in the buffer. After blocking Fc receptors (Human TruStain FcX, 422302, Biolegend, London, UK), 1 µL IL-18 antibody/sample (Human IL-18/IL-1F4 Alexa Fluor 488-conjugated antibody, IC2548G, R&D Systems, Minneapolis, USA) was added and the cells were incubated on ice for 40 min. After 2× washes with perm wash buffer, the cells were incubated with CD14-BV421 (563743, BD), CD56-BUV737 (612767, BD), CD16-APC (561248, BD) antibodies diluted in FACS buffer (PBS with 0.5% BSA) containing Super Bright Complete Staining Buffer (SB-4401-75, eBioscience, Thermo Fisher Scientific, Dartford, UK) for 15 min on ice. After washing 2×, the samples were acquired on a BD Fortessa cell analyser. NK cells were gated as CD16− CD14− CD56+ lymphocytes.

## scRNA-Seq study
Paired admission and follow-up PBMC samples from 5 MIS-C patients were used for 10 × 5' scRNA-Seq. Frozen PBMC were thawed and washed with PBS containing 0.04% BSA, counted, diluted to 1000 cells/µl before being immediately sent to the core scRNA-Seq laboratory. Cells were loaded into a Chromium Controller (10X Genomics) to generate single-cell gel beads in emulsion. Chromium Next GEM Single-Cell 5' HT Kit v2 (PN-1000374, 10X) were used to prepare the gene expression, TCR and BCR libraries, according to the manual. Library fragment size and concentration were assessed with

BioAnalyzer 2100 (Agilent). For sequencing, libraries of 4:1:1 (gene expression:TCR:BCR) were loaded into Illumina NovaSeq flow cells, aiming for >250 M reads/sample.

Sequencing data were aligned and mapped by Cell Ranger (V7.0.1, 10X), with the human reference genome (GRCh38.p14) for gene expression and IMGT database (downloaded on 16 September 2022) for BCR and TCR data. Cell ranger outputs were read into Scanpy package[57]. Dandelion package was used to find cell duplicates[58]. After their removal, the dataset was further filtered by total gene counts <6000/cell. Data were analysed in the Scanpy pipeline. After normalisation and log transformation, highly variable genes were selected for downstream PCA analysis. Samples were integrated by Harmony package[59] to remove the batch effect. UMAP were plotted with the PCA results, with 22 total clusters. The last cluster was dropped out for further analysis as there were less than 20 cells in total. For the remaining clusters, differential genes were calculated for each cluster. To aid cluster annotation, CellTypist package[60] was used to do automatic annotation with Immune_All_Low reference dataset. Each cluster was annotated taking consideration of the gene expression in Leiden clusters and CellTypist results. For detailed CD4 T cell annotation, COVID19_Immune_Landscape reference dataset was used in the CellTypist package. To find a differential abundance of cells between acute and follow-up conditions, the miloR package was used[61]. HLA phenotyping was analysed with scHLAcount[62]. To compare PBMC samples from healthy and MIS-C follow-up children, the dataset from the previous dataset was used for integration[32]. (Package versions: scanpy == 1.9.1 anndata == 0.8.0 umap == 0.5.3 numpy == 1.22.3 scipy == 1.8.1 pandas == 1.4.3 scikit-learn == 1.1.1 statsmodels == 0.13.2 python-igraph == 0.9.11 pynndescent == 0.5.4)

## Statistics & reproducibility

R (version 4.0.3) was used to plot the clinical test results correlation matrix with published code[20], cytokine PCA map with factoextra package (version 1.0.7). OMIQ platform was used to plot the opt-tSNE map. Statistical tests of the quantitative data were performed in GraphPad Prism software (Version 9.4.0, GraphPad Software, California, USA), with detailed information for each figure described in the figure legends. All data points represent biologically independent samples. All 2 group comparisons are conducted in two-sided tests.

No statistics method was used to predetermine the sample size. No data were excluded from the analysis. The experiments were not randomised. The investigators were not blind to allocations during experiments and outcome assessment.

## Reporting summary

Further information on research design is available in the Nature Portfolio Reporting Summary linked to this article.

# Data availability

The scRNA-Seq data generated in this study have been deposited in the Zenodo database under accession code 7997382. The CyToF data from the T cell panel study have been deposited in the FlowRepository database under access code FR-FCM-Z6F8, http://flowrepository.org/id/FR-FCM-Z6F8. Source data are provided with this article. Source data are provided with this paper.

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

## Acknowledgements

We thank all the participants and their families as well as the clinical care and research teams who obtained the samples. This work was funded by Action Medical Research (Registered Charity No. 208701, Grant code: GN2903), Addenbrookes Charitable Trust, and supported by the NIHR Cambridge Biomedical Research Centre (BRC-1215-20014) and the Gates-Cambridge Trust (OPP1144). L.M.D. is supported through the European Union's Horizon 2020 research and innovation programme under the Marie Sklodowska-Curie grant agreement No 955321. Drs. Stephen Baker, Paul Lyons and Rainer Doffinger kindly provided access to Luminex 200 machine. Dr. Richard Grenfell in CRUK Cambridge Institute operated the mass cytometer used to collect CyToF data. Drs. Matthew Lawson Coates and Chuan Xu offered advice on scRNA-Seq data analysis. Dr. Laura Everton in Jeffrey Cheah Biomedical Centre single-cell core facility carried out the scRNA-Seq library preparation. Professor Michael Levin and Dr Jethro Herberg kindly supported the study set-up. The views expressed are those of the authors and not necessarily those of the NIHR or the Department of Health and Social Care.

## Author contributions

Z.Z. designed the experimental protocol, undertook experimental work and data analysis and wrote the manuscript. G.T. designed the CyToF panels and conducted the staining with Z.Z. N.P., C.B. and G.T. conceived and supervised the study, and co-wrote the manuscript. I.R.L.K., J.A.C., E.S., N.Khan, undertook experimental work, proofread the manuscript. L.M.D., K.B.M. and S.A.T. analysed the scRNA-seq data. E.S., E.D., D.W., L.O., C.C., C.P., S.B., K.K., R.G., V.W., H.W., C.R. and K.B. co-ordinated patient enrolment, data and sample collection. N.P., C.B., G.T., P.R., B.S., M.P., N.Klein and H.M. developed the study protocol and obtained research funding.

## Competing interests

In the past three years, S.A.T. has received remuneration for Scientific Advisory Board Membership from Sanofi, GlaxoSmithKline, Foresite Labs and Qiagen. S.A.T. is a co-founder and holds equity in Transition Bio. C.E.B. is on the SAB of NodThera, Lightcast, Related Sciences and Janssen Pharmaceuticals and is a co-founder of Polypharmakos and Danger Bio. N.P. received an honorarium from Biomerieux Diagnostics. The other authors declare no competing interests.

## Additional information

¹Departments of Paediatrics, University of Cambridge, Cambridge, UK. ²Wellcome Sanger Institute, Wellcome Genome Campus, Cambridge, UK. ³Institute of Immunology and Immunotherapy, University of Birmingham, Birmingham, UK. ⁴Paediatric Intensive Care Unit, Addenbrookes Hospital, Cambridge, UK. ⁵Paediatric Intensive Care Unit, Great Ormond Street Hospital, London, UK. ⁶Department of Paediatrics, Imperial College London, London, UK. ⁷Paediatric Intensive Care Unit, Birmingham Children's Hospital, Birmingham, UK. ⁸Institute of Inflammation and Ageing, University of Birmingham, Birmingham, UK. ⁹Departments of Paediatrics, University College London, London, UK. ¹⁰Critical Care, University College London, London, UK. ¹¹Department of Theory of Condensed Matter, Cavendish Laboratory, Department of Physics University of Cambridge, Cambridge, UK. ¹²Department of Medicine, University of Cambridge, Cambridge, UK. ¹³These authors contributed equally: Clare Bryant, Graham Taylor, Nazima Pathan. ✉e-mail: ceb27@cam.ac.uk; g.s.taylor@bham.ac.uk; np409@cam.ac.uk

