## [Peer Review File · Nature Communications]

Enhanced CD95 and Interleukin 18 Signaling Accompany T cell receptor V β 21.3+ Activation in Multi-Inflammatory Syndrome in ChildrenREVIEWER COMMENTS

Reviewer #1 (Remarks to the Author):

In their manuscript, Zhang et al provide high-parameter assessments of mononuclear cells in MIS-C, including extended focus on the cells of most interest, those that are TCRVb21.3 positive. They additionally find changes in the IL18 axis and identify pathway differences from KD that may have clinical relevance. The efforts here are commendable and will serve as a substantial resource for the field with tremendous depth of relevant data. While the approach is robust, the manuscript has some organizational issues that limit a fuller understanding. It occasionally reads as several papers put together. In multiple cases, interesting data features are not mentioned or incompletely explained. Broadly, the cohorts, display symbols, figure order, and figure calls are inconsistent. These issues limited this reviewer's ability to critique the manuscript fully.

- It would be helpful to include clearer information about the subjects:

-Were all non-healthy controls from the ICU?

-Were all MIS-C samples pre-treatment unless otherwise indicated?

-When (time-frame) were samples collected?

-In general, when were samples collected relative to admission?

- It would be helpful for the authors to be clearer about how the following cohort terms relate to one another, as the terms in all figures used do not match those outlined in Fig 1A.

- Figure 1: MIS-C , COVID-19 pneumonia, pediatric control.

- Figure 2: acute MIS-C, COVID-19, noncovid LRTI

- Figures 3 and 4: Healthy, MIS-C acute, MIS-C Follow, COVID, Infection

- Many figures are not discussed or referenced en bloc (e.g. supplemental Figures 7 A-G are mentioned as asides in lines 251 and 267).

- Lines 147-150 require figure calls. It is unclear whether all calls are to Figure 3A, because statistical statements (e.g. 'smaller proportion') are made when no statistics are shown in that figure.

- Line 152 – The supplemental figure calls jump to here from Supp Figure 1 to Supp Figure 5M. This is difficult for the reviewer to follow. 5M then precedes figure calls for SFig5A-D,I,J on line 164). Supplemental Figures 2 and 3 do not appear to be mentioned.

Minor:

- Introduction lines 27 and 28 – suggest omitting the listed overlapping clinical features of MIS-C with KD and TSS as the lists are each incomplete and are not necessary to add clarity.

- Intro line 33: PMID: 33705359 should be referenced for initial discovery of TCRVb gene enrichment in MIS-C. The reference accuracy may need to be checked as the Hoste reference is listed as 2021 but is published in 2022 per PMID: 34914824

- Line 34 – superantigen response comment should be cited, likely referencing Arditi.

- Figure 1– would suggest a different patient symbol for 1A to prevent miscommunication as almost no children included in the studies were infants.

- Line 82: This is surprising, as most covid pneumonia ICU admissions did not occur in young children.

- Results line 88 – PMID: 34893640 and PMID: 33263756 has cytokines from children in the

ICU for COVID-19 – it is unclear whether the authors mean that their distinct data are cytokines in non-covid ICU LRTI.

- Line 110 – Did the authors intend to alternate between IL18BP and IL18BP α ?
- Line 142 – Were all CyTOF markers except TCRV β used in this dimension reduction? If not, would be useful to know which ones used.
- Line 156 needs a figure call to find the data.
- Figure 3B – Symbols are too small even when printed on full sheet of paper to clearly resolve groups.

Reviewer #2 (Remarks to the Author):

Zhang et al. present a multimodal studies on a cohort of patients with MIS-C, acute COVID-19, and non-SARS-CoV-2 lower respiratory tract infections (LRTI). Some of the findings are concordant with what has been previously published (e.g. expansion of CD4 $^{+}$ and CD8 $^{+}$ TCR V β 21.3 $^{+}$ cells, increased IL-18). However, the major limitation of this manuscript is its largely descriptive nature, with statements of causality that are not fully supported by the data presented.

Major comments:

1. The authors acknowledge that the majority of the patients with MIS-C in their cohort were treated with glucocorticoids and IVIG, but (1) do not offer any details in their tables of the cohort to indicate how long patients were treated and how much glucocorticoids they received and (2) do not control for treatment effects. It is therefore not possible to determine which differences are due to treatment effects vs. the pathophysiology of MIS-C itself. For example, treatment with glucocorticoids will blunt the response to LPS, but this is not mentioned or addressed in the authors' discussion of endotoxin tolerance (lines 337 - 340).

2. While the work presented is largely descriptive, the authors present their findings as mechanistic insights. Example of this are as follows:

a. The concurrent expression of CD95 and IL-18R in patients with MIS-C does not demonstrate causality as indicated by this statement: "Unexpectedly, in MIS281 C patients IL-18 appeared to be produced in monocytes via a CD95 mediated." (lines 280 - 282).

b. LBP is an acute phase reactant and is upregulated in response to inflammation, but is not specific for gut epithelial damage as stated in the following: "Gut epithelial damage was proposed as a mechanism of inflammation in MIS-C. Consistent with this hypothesis, we detected increased plasma levels of LBP in our MIS-C patient cohort. LBP is secreted by the liver in response to endotoxin in the blood and increased LBP levels could therefore indicate gut epithelial damage." (lines 286 - 290).

Additional mechanistic studies would be needed to prove the causal relationships mentioned in these statements.

3. In Fig 2V, the authors show that the ratio of IL-18:IL-18BP is low in patients with MIS-C and comparable to that of children with non-SARS-CoV-2 respiratory infections. What is the proof supporting the authors' conclusions that: "Given that IL-18 levels in healthy children are very low, it is likely that the levels of free IL-18 present in MIS-C patients are biologically relevant." (Lines 298 - 300).

4. The authors speculate on the source of increased IL-18 levels in patients with MIS-C, but their functional studies do not identify the source of this cytokine. As IL-18 is central to the authors' pathophysiologic model of MIS-C (lines 368 - 370), the lack of data definitively identifying the source for IL-18 limits the impact of this paper.

Minor comments:

1. The statistical comparison bars on Fig. 2K are incorrect: both are comparing the patients with MIS-C to those with LRTI.

Reviewer #3 (Remarks to the Author):

The authors of this study present a detailed analysis of selected immune parameters in children with MIS-C, compared with children hospitalized for COVID-19 pneumonia and another cohort of children with a severe course of other types of infections. The analysis included a spectrum of serum cytokines, immunophenotyping of T cell subpopulations, complemented by functional studies reflecting NLRP3 inflammasome activation and RNA analysis of selected parameters obtained during the course of the disease.

A fundamental problem with this study arises early in its design in forming patient groups and obtaining samples.

Patient cohorts and cohort groups are insufficiently described. Only basic facts as age, sex and ethnicity are given. The cohort of healthy controls that appear in some experiments is not defined at all. Almost no characteristics on MIS-C clinical variability is presented (although all children were diagnosed according to UK Royal College criteria). Similarly, details are missing for the control group with serious infections that are not specified.

However, most importantly, samples from children in the MIS-C cohort were all obtained after the children had been treated with high doses of immunosuppressive therapy to manage the acute attack of the disease. This treatment is lifesaving and highly significant, involving corticosteroids and high dosed immunoglobulins, among others. The effect of this therapy may be absolutely crucial in view of the experimental design of the study. The focus of this work is to monitor NLRP3 inflammasome activity, which (according to published papers) is very likely to be affected by this treatment. Aware of this limitation, the authors surprisingly state in their commentary that the interval between treatment and sampling in their study was shorter than in other published work, but this implies that their samples must have been under the influence of these medications.

The assumption that children with MIS-C have reduced NLRP3 inflammasome function is therefore debatable, as it is not possible to distinguish between the effect of MIS-C itself or the effect of immunosuppression. The reduced inflammasome function in MIS-C is summarized in Fig. 5, however, according to the data shown here, the reduction is only evident in IL-1 production detected in stimulated samples, whereas IL-18 secretion is higher in the aforementioned cohort of MIS-C patients. The interpretation of the results seems misleading, the data suggesting rather a switch of NLRP3 inflammasome activation to IL-18 dominance (similar situations are described in the literature).

The authors also looked at the production of IL-18 itself, which appears to be based on many cell types, dominated by monocytes and NK cells in MIS-C. On this point, the authors suggest that these cells produce IL-18 through a noncanonical CD95-mediated process, but they show only circumstantial evidence for this.

Thus, the conclusions drawn from the part of the manuscript concerning IL-18 production

and reduced inflammasome function in children with MIS-C are somewhat confusing and would require further specification.

The next part of the paper deals with the clonality of T lymphocytes in the MIS-C cohort. This aspect has already been repeatedly described, compared to previous works the authors describe increased expression of ICOS, CD28 and IL18R markers on these cells and direct their appearance to Th1, Th17 and Treg compartments. This is a part of the paper not directly related to the NLRP3 inflammasome and it is questionable whether to combine these parts into one communication.

Overall, this is a communication that complements the already known facts about MIS-C and highlights the potentially more important role of IL-18 in the disease. However, it appears that more detailed evidence would be needed to support its conclusions.

Reviewer #4 (Remarks to the Author):

Zhang and colleagues analyze blood samples from children with MIS-C, acute respiratory SARS-CoV-2, and non SARS-CoV-2 related infections. They use deep immune profiling, transcriptional and in vitro stimulations to explore these responses.

The analyses are well executed and the conclusions are well supported. I had a hard time understanding how the single cell transcriptomics extends the narrative and think the authors could have woven this in a bit better.

REVIEWER COMMENTS

Reviewer #1 (Remarks to the Author):

In their manuscript, Zhang et al provide high-parameter assessments of mononuclear cells in MIS-C, including extended focus on the cells of most interest, those that are TCRVb21.3 positive. They additionally find changes in the IL18 axis and identify pathway differences from KD that may have clinical relevance. The efforts here are commendable and will serve as a substantial resource for the field with tremendous depth of relevant data. While the approach is robust, the manuscript has some organizational issues that limit a fuller understanding. It occasionally reads as several papers put together. In multiple cases, interesting data features are not mentioned or incompletely explained. Broadly, the cohorts, display symbols, figure order, and figure calls are inconsistent. These issues limited this reviewer's ability to critique the manuscript fully.

We appreciate the reviewer's comments here and have addressed the organisation, continuity of flow and data features to read better and be more informative to the reader. Further, we have proofread the manuscript carefully to ensure the cohorts, figure order and symbols are consistent.

- It would be helpful to include clearer information about the subjects:

-Were all non-healthy controls from the ICU?

We now provide this information in a newly added supplementary file 1. Non-healthy controls are drawn from ward and ICU. The non-covid lower respiratory tract infection cohort are all drawn from the ICU. We have included a range of controls due to the mixed and changing nature of disease presentation in children during successive waves of the COVID-19 pandemic.

-Were all MIS-C samples pre-treatment unless otherwise indicated?

No, there are 23 samples without pre-treatment without IVIG or glucocorticoids in 43 total MIS-C samples. We now include the pre-treatment information of MIS-C samples in whole blood functional assay in the supplementary file 1. In this assay, 3 out 14 MIS-C samples were without pre-treatment with glucocorticoids/IVIG.

Unfortunately, the logistics of critical care in the UK (where children are transported from a network of general paediatric wards if they require a high intensity of nursing/medical care) required us to make a pragmatic decision to include all children. Some of them may have received treatment at the receiving hospital (steroids or occasionally IVIG) which we recorded. We have added a table of the timeframe of treatment and sample collection, as well as admission time. The information also covers the 2 points below.

-When (time-frame) were samples collected?

For MIS-C, samples at admission (usually within 1 day) were collected for all the study subjects. Where vascular access was possible, we also collected samples after treatment (2-4 days after admission), at discharge (about 7 days after admission) and follow-up (about 1 month post discharge) stages.

-In general, when were samples collected relative to admission?

The first samples were all collected within about 24 hours of admission to PICU or to the ward. This information is in the supplementary file 1.

- It would be helpful for the authors to be clearer about how the following cohort terms relate to one another, as the terms in all figures used do not match those outlined in Fig 1A.

- Figure 1: MIS-C , COVID-19 pneumonia, pediatric control.

- Figure 2: acute MIS-C, COVID-19, noncovid LRTI

- Figures 3 and 4: Healthy, MIS-C acute, MIS-C Follow, COVID, Infection

We have standardised the classification of patient cohorts into three groups - MIS-C, COVID-19 pneumonia and Non-Covid. We hope the inclusion of additional clinical details and sample timing helps bring more clarity.

We have revised the manuscript, including introductory Figure 1 to reflect the changes, including description of the healthy control cohort.

- Many figures are not discussed or referenced en bloc (e.g. supplemental Figures 7 A-G are mentioned as asides in lines 251 and 267).

We apologise for the oversight. In trying to bring across the breadth of work, some clarity has been lost in attempting to maintain brevity of words. We have revised our manuscript including the figures used and ensuring individual figures, where used, are fully referenced to in the main text.

- Lines 147-150 require figure calls. It is unclear whether all calls are to Figure 3A, because statistical statements (e.g. 'smaller proportion') are made when no statistics are shown in that figure.

We have amended to manuscript to improve clarity on this point:

'In contrast, levels of IL-18R and the co-stimulation receptors CD28 and ICOS were more prominent in TCR Vβ21.3+ CD4+ T cells than CD8+ T cells (Figure 3A, 4th, 5th and 6th panel). A -small proportion of TCR Vβ21.3+ CD8 and CD4 T-cells expressed immune checkpoint markers PD-1 and CD39 (Figure 3A, 7th and 8th panel).'

We have removed statistical statements in this part as these lines describes all about Figure 3A. Given that we undertook detailed statistical tests in the manual gating process, we did not carry out additional statistical test with the tSNE plots.

- Line 152 – The supplemental figure calls jump to here from Supp Figure 1 to Supp Figure 5M. This is difficult for the reviewer to follow. 5M then precedes figure calls for SFig5A-D,I,J on line 164). Supplemental Figures 2 and 3 do not appear to be mentioned.

We have added text explaining each of the supplementary files and corrected any discontinuity in file order.

'Data cleaning and unmixing was shown in Supplementary Figure 2. Manual gating in Flowjo software was shown in Supplementary Figure 3. Exploratory analysis results in R were shown in Supplementary Figure 4, which suggested changes between MIS-C and other groups in cell

populations (Supplementary Figure 4A, B), as well as levels of cell surface markers like CD95 and IL-18R (Supplementary Figure 4C, D).'

Minor:

- Introduction lines 27 and 28 – suggest omitting the listed overlapping clinical features of MIS-C with KD and TSS as the lists are each incomplete and are not necessary to add clarity.

We agree and have removed those lists.

- Intro line 33: PMID: 33705359 should be referenced for initial discovery of TCRVb gene enrichment in MIS-C. The reference accuracy may need to be checked as the Hoste reference is listed as 2021 but is published in 2022 per PMID: 34914824

We have added the suggested reference. In terms of the Hoste paper, we double checked and it is right to cite it in 2021. The paper was first published online in Dec 2021. For this reason, we have kept the original citation but are happy to be advised by the editorial team.

- Line 34 – superantigen response comment should be cited, likely referencing Arditi.

We have added this reference, thank you.

- Figure 1– would suggest a different patient symbol for 1A to prevent miscommunication as almost no children included in the studies were infants.

We agree and have amended the figure.

- Line 82: This is surprising, as most covid pneumonia ICU admissions did not occur in young children.

We have added explanation that these patients only emerged in the omicron wave when respiratory presentations became more prevalent and COVID-related pneumonia was a cause of hospital and PICU admission.

- Results line 88 – PMID: 34893640 and PMID: 33263756 has cytokines from children in the ICU for COVID-19 – it is unclear whether the authors mean that their distinct data are cytokines in noncovid ICU LRTI.

Yes, the distinction is our comparison of MIS-C with a group of children with severe respiratory infection.

- Line 110 – Did the authors intend to alternate between IL18BP and IL18BPa?

We clarified this and use IL-18BP in the text, '*Plasma IL-18 activity is regulated by the neutralising protein IL-18 binding proteins (IL-18 BP, with IL-18 BPa as the main splice variant).*'

- Line 142 – Were all CyTOF markers except TCRVb used in this dimension reduction? If not, would be useful to know which ones used.

No, we used markers in T cell panel except for CD45, B2M, CD14, CD56, CD19, TCR-Vβ 21.3.

- Line 156 needs a figure call to find the data.

We now added a tSNE plot breaking the infection groups into sepsis and non-sepsis samples, supplementary figure 5B.

- Figure 3B – Symbols are too small even when printed on full sheet of paper to clearly resolve groups.

We increased the symbol sizes.

Reviewer #2 (Remarks to the Author):

Zhang et al. present a multimodal studies on a cohort of patients with MIS-C, acute COVID-19, and non-SARS-CoV-2 lower respiratory tract infections (LRTI). Some of the findings are concordant with what has been previously published (e.g. expansion of CD4+ and CD8+ TCR V β 21.3+ cells, increased IL-18). However, the major limitation of this manuscript is its largely descriptive nature, with statements of causality that are not fully supported by the data presented.

We agree our data demonstrates validity through concurrence with previously published observations, but aspects of our work provide advancements on knowledge of MIS-C pathophysiology in key areas. In terms of IL-18, previous study on it in MIS-C is only limited to its cytokine levels. We found out the IL-18R is highly increased in those TCR Vb21.3 T cells and NK cells, and explored mechanisms of its production and cell sources. In terms of TCR Vb21.3 T cells, previous researchers have focused on CD8 compartment, not mentioning involvement of co-stimulation receptors (such as CD28, ICOS, IL-18R). The CD4 compartment has been largely overlooked, with no phenotyping work on them. Nor has any previous work studied their differentiation states, such as Th1 and Treg. Our discovery that TCR Vb21.3+ CD4+ T cells are enriched not only in Th1, but also immune suppressive Treg state will be very interesting to people in the field.

We agree the lack of mechanistic insight was a limitation of the original manuscript and have carried out additional work to explore the disease mechanism. For example, in terms of IL-18 production, we did ASC staining (to look for ASC specks, marker of NLRP3 inflammasome activation) to further examine the NLRP3 activation state in immune cells from children with MIS-C. After finding few ASC positive cells in MIS-C blood samples, we explored CD95 (FAS) downstream signalling target, caspase 8, which was reported to activate IL-18 production (PMC3518757). In agreement with our prediction that caspase 8 might be driving IL18 production, we found enhanced caspase 8 activity in MIS-C monocytes from recovered PBMC samples. The data highlight monocyte CD95 and caspase 8 activation in MIS-C, which could lead to high IL-18 production

In addition, we carried out intracellular staining of IL-18 in fixed blood samples and confirmed MIS-C CD56+ NK cells had high IL-18 at protein level as well.

Major comments:

The authors acknowledge that the majority of the patients with MIS-C in their cohort were treated with glucocorticoids and IVIG, but do not offer any details in their tables of the cohort to indicate how long patients were treated and how much glucocorticoids they received

We have added supplementary file 1 detailing the timeframe of treatment and sampling. The treatment effect is now given in more detail.

They do not control for treatment effects. It is therefore not possible to determine which differences are due to treatment effects vs. the pathophysiology of MIS-C itself. For example, treatment with glucocorticoids will blunt the response to LPS, but this is not mentioned or addressed in the authors' discussion of endotoxin tolerance (lines 337 - 340).

We fully understand the concerns. We agree this is a limitation imposed by the need for a pragmatic and inclusive study during a fast changing pandemic admission cohort. To give further context in the revised manuscript, we undertook further analysis and have added clarifications and limitations.

First, we agree that the reduced NLRP3 in MIS-C admission samples may be influenced by treatment such as glucocorticoids/IVIG and we now state this explicitly in the manuscript. Also, for clarity, we have marked those samples from children with MIS-C obtained before treatment had been given (revised Figure 5). Secondly, instead of functional readout, we examined an NLRP3 activation marker, ASC specks in fixed blood samples. There were few ASC speck positive cells in MIS-C. Thirdly, we examined NLRP3 response in MIS-C follow-up samples (which are free from treatments), in comparison to the controls. There was no increased NLRP3 response. In conclusion, there is no evidence of increased NLRP3 activation in MIS-C.

More importantly, we realise the criticism around treatment mainly due to our poor explanation of the aims of our study. The main aim of the paper is to answer whether increased NLRP3 activity underlies MIS-C development, explaining the clinical observations related to fever and cardiovascular involvement. We are not trying to advocate a deficient NLRP3 function in MIS-C. We agree that original statement of 'reduced NLRP3 function in MIS-C' is misleading and thus corrected this as 'no evidence of overactive NLRP3 activity in MIS-C'. We believe this reflects well the findings and is an important message for MIS-C researchers.

2. While the work presented is largely descriptive, the authors present their findings as mechanistic insights. Example of this are as follows:

a. The concurrent expression of CD95 and IL-18R in patients with MIS-C does not demonstrate causality as indicated by this statement: "Unexpectedly, in MIS281 C patients IL-18 appeared to be produced in monocytes via a CD95 mediated." (lines 280 - 282).

We did more work and found higher caspase 8 levels in MIS-C monocytes, which is downstream of CD95 and reported can activate IL-18 production.

The revised manuscript has added detail: 'Unexpectedly, in MIS-C patients IL-18 doesn't appear to be produced by overactive NLRP3 inflammasome pathway. Instead, we see high CD95 levels in monocytes and the downstream caspase 8 activity, which could activate IL-18 production. High IL18 transcript levels and proteins were found in CD16- NK cells in MIS-C.'

b. LBP is an acute phase reactant and is upregulated in response to inflammation, but is not specific for gut epithelial damage as stated in the following: "Gut epithelial damage was proposed as a mechanism of inflammation in MIS-C. Consistent with this hypothesis, we detected increased plasma levels of LBP in our MIS-C patient cohort. LBP is secreted by the liver in response to endotoxin in the blood and increased LBP levels could therefore indicate gut epithelial damage." (lines 286 - 290).

Additional mechanistic studies would be needed to prove the causal relationships mentioned in these statements.

We agree that LBP increase it not specific to gut epithelium damage. And thus, we have corrected our discussion as below.

‘Consistent with the study, we detected increased plasma levels of LBP in our MIS-C patient cohort. LBP is secreted by the liver in response to acute stress, like increased endotoxin in the blood.’

In our paper, we didn’t explore gut damage in MIS-C. It is only mentioned as a claim from literature. As it is not very relevant to this work, we didn’t include more mechanism study here. However, we agree the importance of gut involvement in MIS-C and have another manuscript on this topic.

3. In Fig 2V, the authors show that the ratio of IL-18:IL-18BP is low in patients with MIS-C and comparable to that of children with non-SARS-CoV-2 respiratory infections. What is the proof supporting the authors' conclusions that: "Given that IL-18 levels in healthy children are very low, it is likely that the levels of free IL-18 present in MIS-C patients are biologically relevant." (Lines 298 - 300).

Free IL-18 levels in healthy people are debatable. IL18 has been reported to be undetectable in previous work, PMID: 29326099, but given there is insufficient previous data to be conclusive, we agree this could be controversial and have removed this sentence.

4. The authors speculate on the source of increased IL-18 levels in patients with MIS-C, but their functional studies do not identify the source of this cytokine. As IL-18 is central to the authors' pathophysiologic model of MIS-C (lines 368 - 370), the lack of data definitively identifying the source for IL-18 limits the impact of this paper.

To explore the source of IL-18 production, we performed experiments to explore CD95 (FAS) signalling, which has been linked to caspase 8, and has been reported to drive IL-18 activation. In agreement with our prediction, higher caspase 8 activity is found in MIS-C monocytes from recovered PBMC samples. In addition, we carried out intracellular staining of IL-18 in fixed blood samples and confirmed MIS-C CD56+ NK cells had high IL-18 at protein level as well.

Minor comments:

1. The statistical comparison bars on Fig. 2K are incorrect: both are comparing the patients with MIS-C to those with LRTI.

Apologies, there is a misunderstanding here. In figure 2K, open line represents ANOVA test and capped line represents post-hoc test.

Reviewer #3 (Remarks to the Author):

The authors of this study present a detailed analysis of selected immune parameters in children with MIS-C, compared with children hospitalized for COVID-19 pneumonia and another cohort of children with a severe course of other types of infections. The analysis included a spectrum of serum cytokines, immunophenotyping of T cell subpopulations, complemented by functional studies

reflecting NLRP3 inflammasome activation and RNA analysis of selected parameters obtained during the course of the disease.

A fundamental problem with this study arises early in its design in forming patient groups and obtaining samples.

Patient cohorts and cohort groups are insufficiently described. Only basic facts as age, sex and ethnicity are given. The cohort of healthy controls that appear in some experiments is not defined at all. Almost no characteristics on MIS-C clinical variability is presented (although all children were diagnosed according to UK Royal College criteria). Similarly, details are missing for the control group with serious infections that are not specified.

Apologizes for this. We have added further information about the healthy controls in the supplementary file 1, as well as information about the none COVID-19 related control patients in supplementary table 1. For MIS-C patients, we added information about their clinical variability and a new table detailing the timeframe of treatment and sampling in the same file. We have provided as much detail as we can, but our ethical approval and the need to respect patient confidentiality provides a limit to the granularity of such data.

However, most importantly, samples from children in the MIS-C cohort were all obtained after the children had been treated with high doses of immunosuppressive therapy to manage the acute attack of the disease. This treatment is lifesaving and highly significant, involving corticosteroids and high dosed immunoglobulins, among others. The effect of this therapy may be absolutely crucial in view of the experimental design of the study. The focus of this work is to monitor NLRP3 inflammasome activity, which (according to published papers) is very likely to be affected by this treatment. Aware of this limitation, the authors surprisingly state in their commentary that the interval between treatment and sampling in their study was shorter than in other published work, but this implies that their samples must have been under the influence of these medications.

This is a very reasonable point. We believe it is more meaningful to analyse acutely ill patients as soon as possible, rather than waiting for any potential drug-induced effects to subside, since in this case it is not the acute disease that is being studied. Again, we agree that this also means our functional assay may be influenced by treatments.

Bearing in mind of these concerns, we tried to analyse the effect of pre-treatment on NLRP3 activation from our data and in literature. The finding is a bit surprising: NLRP3 pathway is not effectively suppressed by immune suppression drugs like IVIG or steroid.

In our scRNA-seq study, there are 2 patients whose admission samples were collected before IVIG/steroids treatment, 1 pretreated with IVIG and 2 pretreated with GC. When gene expression in monocytes were compared, neither IL1B, IL18 or NLRP3 were different between GC group and no treatment, or between IVIG group and no treatment although we appreciate this is a very limited number of subjects.

This piece of information is interesting as in previous scRNA-seq study of KD patients, at least IL1B was found to be suppressed by treatment (PMID: 34521850). However, in MIS-C, the effect on IL-1 β is less clear. In MIS-C study of treatment effects, neither IL-1 β or IL-18 levels were affected by IVIG or glucocorticoid treatment (Figure 5, PMC7474869). This is in agreement with our data (Supplementary Figure 1G).

To be transparent about this for reader, in our whole blood functional assay, we now labelled the 3 MIS-C samples without pre-treatment with glucocorticoids/IVIG (revised Figure 5). There is no clear pattern of treatment effect in the data spread in MIS-C group.

Instead of only replying on NLRP3 functional assay, we went further to detect NLRP3 activation marker ASC specks in fixed MIS-C blood samples, there is no evidence of increased ASC speck staining, comparing MIS-C admission to discharge samples, or compared to positive control of LPS plus ATP treated blood samples.

In addition, we compared MIS-C NLRP3 function at follow-up stage when there is little treatment effect influence, to paediatric patient samples. There was no overactive NLRP3 activity in children with MIS-C.

In conclusion the literature on the effects of immune suppression reagents on NLRP3 activation is highly variable, and our data do not show clear evidence of suppression of NLRP3 pathway by the pre-treatments. We have now modified the text to explain the potential complications of immunosuppressive therapy and interpretation of our data, and did more analysis related to NLRP3 pathway in MIS-C.

The assumption that children with MIS-C have reduced NLRP3 inflammasome function is therefore debatable, as it is not possible to distinguish between the effect of MIS-C itself or the effect of immunosuppression. The reduced inflammasome function in MIS-C is summarized in Fig. 5, however, according to the data shown here, the reduction is only evident in IL-1 production detected in stimulated samples, whereas IL-18 secretion is higher in the aforementioned cohort of MIS-C patients. The interpretation of the results seems misleading, the data suggesting rather a switch of NLRP3 inflammasome activation to IL-18 dominance (similar situations are described in the literature).

We agree that 'MIS-C have reduced NLRP3' is debatable and thus we reworded it as 'there is no overactive NLRP3 in MIS-C'.

We thank the reviewer for kindly highlighting high IL-18 levels in MIS-C group. As its levels don't change upon stimulation in MIS-C group, we don't think there is active secretion in MIS-C group in our in vitro experiments and thus no switch to IL-18 dominance.

One possible explanation for this observation includes more stability of IL-18 compared to IL-1beta in vivo. In sepsis, IL-18 remained high from day 1 to day 7, while IL-1beta peaked in day 1 and returned to normal in day 7 (PMCID: PMC7889521).

The authors also looked at the production of IL-18 itself, which appears to be based on many cell types, dominated by monocytes and NK cells in MIS-C. On this point, the authors suggest that these cells produce IL-18 through a noncanonical CD95-mediated process, but they show only circumstantial evidence for this.

We agree that ‘the production of IL-18 itself, which appears to be based on many cell types, dominated by monocytes and NK cells in MIS-C’. So we changed the text as below.

‘While IL1B expression was confined to monocytes, IL18 expression is more ambiguous. Nevertheless, it is clear that CD16- NK cells also express high IL18 in MIS-C, in addition to monocytes ((Figure 6E).’

For monocytes, we did caspase 8 (downstream of CD95 and upstream of IL-18) activity assays and found much higher activity in MIS-C than COVID-19 samples (revised Figure 5C). Also, we did IL-18 cytokine staining in fixed blood cells and found high levels of it in monocytes and CD56+ NK cells in MIS-C (revised Figure 6G).

Thus, the conclusions drawn from the part of the manuscript concerning IL-18 production and reduced inflammasome function in children with MIS-C are somewhat confusing and would require further specification.

To address this point, we carried out more work related to NLRP3 marker ASC speck, monocyte caspase 8 activity assay and intracellular IL-18 cytokine assay.

We modified the statement of reduced NLRP3 activity in MIS-C into ‘there is no evidence of inherently overactive NLRP3 activity’.

The next part of the paper deals with the clonality of T lymphocytes in the MIS-C cohort. This aspect has already been repeatedly described, compared to previous works the authors describe increased expression of ICOS, CD28 and IL18R markers on these cells and direct their appearance to Th1, Th17 and Treg compartments. This is a part of the paper not directly related to the NLRP3 inflammasome and it is questionable whether to combine these parts into one communication.

Given the importance of IL-18 and IFN-gamma signature in MIS-C, we think role of T cell and NK cells (major producer of IFN-gamma) being brought together are a strength of this manuscript. The data on immune cell activation through data on T cell activation is important and as the reviewer points out, this aspect offers novel insights of the T cell markers in MIS-C, and provided detailed T helper cell characterisation. All these are new to understand how superantigen activates T cells in MIS-C. In addition, most previous authors focused on the role of CD8+ TCR Vbeta 21.3+ cells. Our work provided detailed analysis on the contribution of CD4+ cells and found surface markers in CD4 T cells are much better than CD8 T cells for diagnosis of MIS-C.

Overall, this is a communication that complements the already known facts about MIS-C and highlights the potentially more important role of IL-18 in the disease. However, it appears that more detailed evidence would be needed to support its conclusions.

We cannot find evidence that explores the activity of the NLRP3 inflammasome in MIS-C patients before and thus these data are new. In terms of T cells, highlighting the IL-18 link in MIS-C is very helpful to understand the disease mechanism.

Our T cell works are novel. We apologise this is not well introduced in the beginning. So far, there have been no high dimensional studies directly on TCR Vb21.3+ T cells. All previous works only focused on CD8+ T cells and the CD4+ compartment has been overlooked. In our paper, we clearly showed there are a large number of TCR Vb21.3+ CD4+ T cells in MIS-C, co-expressing lots of stimulation markers. It is novel that they not only displayed Th1 but also Treg phenotypes. Our work also shows for the first time how T cell activation in MIS-C compares to T cells in children with sepsis, showing a great similarity. In addition, we found activation markers in TCR Vb21.3+ CD4+ T cells, rather the widely reported CD8+ T cells are characteristic of the inflammatory profile seen at admission in MIS-C.

With your constructive advice, we provided more detailed evidence in the revised manuscript.

Reviewer #4 (Remarks to the Author):

Zhang and colleagues analyze blood samples from children with MIS-C, acute respiratory SARS-CoV-2, and non SARS-CoV-2 related infections. They use deep immune profiling, transcriptional and in vitro stimulations to explore these responses.

The analyses are well executed and the conclusions are well supported. I had a hard time understanding how the single cell transcriptomics extends the narrative and think the authors could have woven this in a bit better.

Apologies for the organisation of this part in the original manuscript. We have made lots of changes in describing the scRNA-seq results and hope this reads better in the revised version.

REVIEWER COMMENTS

Reviewer #2 (Remarks to the Author):

The authors have done much extra work and addressed the reviewers' comments well.

Editorial note: This reviewer was also asked to comment on the authors response to reviewer 1 who was unavailable to comment this round of review

"I have re-reviewed the revised manuscript and author responses to Reviewer 1. I feel that the revision has adequately addressed the comments from reviewer 1."

Reviewer #3 (Remarks to the Author):

The authors responded to all comments and significantly improved the manuscript. But it still seems that 2 fundamental issues that are not sufficiently substantiated, the absence of NLRP3 activation (contrary to previously published data) and the expansion V β 21.3+ T on T regs.

T cell analysis

♣ This sentence in the Introduction does not seem to be entirely correct -

These studies have focused on cytotoxic CD8+ T cells and CD4 + T cells have not been characterised -

Here are at least 3 articles that deal with the issue, the statement above should be worded more carefully and emphasize more detailed analysis provided here.

Moreews, M., Le Gouge, K., Khaldi-Plassart, S., et al. Polyclonal expansion of TCR Vbeta 21.3+ CD4+ and CD8+ T cells is a hallmark of Multisystem Inflammatory Syndrome in Children. *Sci Immunol.* 2021;6(59):eabh1516.

Zhang, D., Zhou, Y., Chen, H., et al. T Cells in Multisystem Inflammatory Syndrome in Children (MIS-C) Have a Predominant CD4+ T Helper Response to SARS-CoV-2 Peptides and Numerous Virus-Specific CD4- CD8- Double-Negative T Cells. *J Immunol.* 2022;208(11):2479-2489.

Ramaswamy A, Brodsky NN, Sumida TS, Comi M, Asashima H, Hoehn KB, Li N, Liu Y, Shah A, Ravindra NG, Bishai J, Khan A, Lau W, Sellers B, Bansal N, Guerrerio P, Unterman A, Habet V, Rice AJ, Catanzaro J, Chandnani H, Lopez M, Kaminski N, Dela Cruz CS, Tsang JS, Wang Z, Yan X, Kleinstein SH, van Dijk D, Pierce RW, Hafler DA, Lucas CL. Immune dysregulation and autoreactivity correlate with disease severity in SARS-CoV-2-associated multisystem inflammatory syndrome in children. *Immunity.* 2021 May 11;54(5):1083-1095.e7

♣ In CD4+ T cells, we found expanded V β 21.3+ T cells into the Th1, Th17 and Treg compartment. This is the first time that increased TCR V β 21.3+ Treg cells were reported in MIS-C. I have trouble to find Figures for this statement, Suppl. Fig.5 does not show Tregs or Th17 with V β 21.3 expansion?

NLRP3 activation

We provide detailed characterisation of the expanded TCR V β 21.3+ T cells together with NK cells and monocytes in MIS-C, with no evidence of an overactive NLRP3 inflammasome

pathway but increased apoptosis related caspase 8 signalling in the monocytes. The issue of NLRP3 was addressed before, for example here with different conclusions, the NLRP3 activation status should be discussed with regard to previously published references and in line with presented results.

Konstantinos I. Papadopoulos, Alexandra Papadopoulou, Tar-Choon Aw. A protective erythropoietin evolutionary landscape, NLRP3 inflammasome regulation, and multisystem inflammatory syndrome in children, *Hum Cell*. 2023; 36(1): 26–40.

Wang WT, He M, Shimizu C, Croker BA, Hoffman HM, Tremoulet AH, Burns JC, Shyy JY. Inflammasome Activation in Children With Kawasaki Disease and Multisystem Inflammatory Syndrome.

Arterioscler Thromb Vasc Biol. 2021 Sep;41(9):2509-2511

Results

Also in this context, data on Fig.5 are confusing. On page 11, the authors state As expected, addition of LPS plus ATP induced high levels of IL-1 β in the blood of non-MIS-C patients (paediatric admission and COVID-19 pneumonia) at both the 4h and 24h time points (Figure 5A). LPS plus ATP also increased levels of IL-18 in 254 blood samples from the paediatric admission patients, although only at the 24h time point (Figure 5B). However, for MIS-C patients, LPS plus ATP induced a much smaller increase in IL-1 β and IL-18 levels were unchanged.

The authors, however, do not comment on high levels of IL-1 β in MIS-C follow-up group (Fig.5A)? Again, is there a possibility that IL-1 β levels in acute MIS-C were suppressed by intensive therapy?

The evidence for lack of activation of NLRP3 in MIS-C patients is also not convincing. Fig.6 F-H, showing ASC, is not very well visible and does not include a proper control – only MIS-C sample in acute and follow-up stages. Considering that Fig.5 shows high levels of IL-1 β in MIS-C follow-up samples, there still might be some activation of NLRP3 ongoing?

Overall, the title,

NLRP3 independent IL-18 signaling orchestrates TCR V β 21.3+ T cell activation in acute multisystem inflammatory syndrome in children (MIS-C)

seems too strong and supported only indirectly by presented data.

Reviewer #4 (Remarks to the Author):

The authors have addressed my concerns.

RESPONSE TO REVIEWERS

Manuscript ref: NCOMMS-22-44339C

We are grateful to the journal for the opportunity to respond to the points raised by reviewer 3 and below is a detailed, point by point response with the changes we made highlighted in red below and in the revised manuscript. In addition, we have added additional commentary (in red) to our previous comments (in blue) in response to the previous feedback from reviewer 3 to ensure any and all areas of concern have been addressed.

Reviewer #2 (Remarks to the Author):

The authors have done much extra work and addressed the reviewers' comments well.

*****Editorial note: This reviewer was also asked to comment on the authors response to reviewer 1 who was unavailable to comment this round of review*****

"I have re-reviewed the revised manuscript and author responses to Reviewer 1. I feel that the revision has adequately addressed the comments from reviewer 1."

Thank you for your comments.

Reviewer #4 (Remarks to the Author):

The authors have addressed my concerns.

Thank you for evaluating the manuscript.

Reviewer #3 (Remarks to the Author):

The authors responded to all comments and significantly improved the manuscript. But it still seems that 2 fundamental issues that are not sufficiently substantiated, the absence of NLRP3 activation (contrary to previously published data) and the expansion V β 21.3+ T on T regs.

T cell analysis

This sentence in the Introduction does not seem to be entirely correct -

These studies have focused on cytotoxic CD8+ T cells and CD4 + T cells have not been characterised -

Here are at least 3 articles that deal with the issue, the statement above should be worded more carefully and emphasize more detailed analysis provided here.

Moreews, M., Le Gouge, K., Khaldi-Plassart, S., et al. Polyclonal expansion of TCR Vbeta 21.3+ CD4+ and CD8+ T cells is a hallmark of Multisystem Inflammatory Syndrome in Children. *Sci Immunol.* 2021;6(59):eabh1516.

Zhang, D., Zhou, Y., Chen, H., et al. T Cells in Multisystem Inflammatory Syndrome in Children (MIS-C) Have a Predominant CD4+ T Helper Response to SARS-CoV-2 Peptides and Numerous Virus-Specific CD4- CD8- Double-Negative T Cells. *J Immunol.* 2022;208(11):2479-2489.

Ramaswamy A, Brodsky NN, Sumida TS, Comi M, Asashima H, Hoehn KB, Li N, Liu Y, Shah A, Ravindra NG, Bishai J, Khan A, Lau W, Sellers B, Bansal N, Guerrero P, Unterman A, Habet V, Rice AJ, Catanzaro J, Chandnani H, Lopez M, Kaminski N, Dela Cruz CS, Tsang JS, Wang Z, Yan X, Kleinstein SH, van Dijk D, Pierce RW, Hafler DA, Lucas CL. Immune dysregulation and autoreactivity correlate with disease severity in SARS-CoV-2-associated multisystem inflammatory syndrome in children. *Immunity.* 2021 May 11;54(5):1083-1095.e7

We have included the references suggested by the referee to include these references. We have amended the introduction to reflect prior work on CD4 T cells in MIS-C as shown below.

There appear to be immunological similarities between MIS-C and TSS. Increased frequencies of TCR Vβ21.3+ (gene name: TRBV11-2) positive T-cells are observed in both CD4+ and CD8+ T cells in MIS-C²⁻⁵, which resembles superantigen activation of specific Vβ subsets of T cells (e.g. TSS toxin-1 that targets Vβ-2 T cells)^{6,7}. In MIS-C a variety of cytokines are increased in plasma, including Th1 type IFN-γ and downstream chemokines CXCL9 and CXCL10², as seen in TSS⁸. However, while CD4+ T cells have a clear dominant role in TSS development⁹, data on CD4+ T cells in MIS-C are more limited¹⁰, particularly in relation to T helper (Th) cell differentiation which has not been fully explored.

In CD4+ T cells, we found expanded Vβ21.3+ T cells into the Th1, Th17 and Treg compartment. This is the first time that increased TCR Vβ21.3+ Treg cells were reported in MIS-C.

I have trouble to find Figures for this statement, Suppl. Fig.5 does not show Tregs or Th17with Vβ21.3 expansion?

We apologise for the lack of clarity in this sentence. The data are shown in Figure 3D (as below). We realise this panel is too crowded and thus we have split them up in the revised figure. In addition, the position of the subheading (Expanded CD4+ TCR Vβ21.3+ cells in MIS-C patients consist of Th1, Th17, Treg, but not Th2 cells) is adjusted to close to the related results.

Figure 3D Percentage of TCR Vβ21.3+ cells in CD8 T-cells (divided by differentiation state, based on CD45RA and CD27 levels) or CD4+ T-cells (divided into Th1, Th2, Th17 and Treg subsets based on chemokine receptor expression). Kruskal-Wallis statistic test was used to compare the five groups, with Dunn's multiple comparisons test to compare each group to the MIS-C group. Naïve T cells were CD45RA+ CD27+, central memory (CM) CD45RA- CD27+, effector memory (EM) CD45RA- CD27- and terminal effector (TE) CD45RA+ CD27.

Figure 3 (D) Percentage of TCR Vβ21.3+ cells in CD8 T-cells (divided by differentiation state, based on CD45RA and CD27 levels) subsets. Kruskal-Wallis statistic test was used to compare the five groups, with Dunn's multiple comparisons test to compare each group to the MIS-C group. Naïve T cells were CD45RA+ CD27+, central memory (CM) CD45RA- CD27+, effector memory (EM) CD45RA- CD27- and terminal effector (TE) CD45RA+ CD27. (E) Percentage of TCR Vβ21.3+ cells in CD4+ T-cells (divided into Th1, Th2, Th17 and Treg subsets based on chemokine receptor expression). Kruskal-Wallis statistic test was used to compare the five groups, with Dunn's multiple comparisons test to compare each group to the MIS-C group.

NLRP3 activation

We provide detailed characterisation of the expanded TCR Vβ21.3+ T cells together with NK cells and monocytes in MIS-C, with no evidence of an overactive NLRP3 inflammasome pathway but increased apoptosis related caspase 8 signalling in the monocytes.

We have amended this introductory sentence as below.

'We provide detailed characterisation of the expanded TCR Vβ21.3+ T cells together with NK cells and monocytes in MIS-C. No overactive NLRP3 inflammasome pathway activity could be seen, but increased FAS expression and active caspase 8 was seen in the monocytes and it is possible this could act as an alternative processor of inflammasome substrates in MIS-C patient.'

The issue of NLRP3 was addressed before, for example here with different conclusions, the NLRP3 activation status should be discussed with regard to previously published references and in line with presented results.

Konstantinos I. Papadopoulos, Alexandra Papadopoulou, Tar-Choon Aw. A protective erythropoietin evolutionary landscape, NLRP3 inflammasome regulation, and multisystem inflammatory syndrome in children, *Hum Cell.* 2023; 36(1): 26–40.

Wang WT, He M, Shimizu C, Croker BA, Hoffman HM, Tremoulet AH, Burns JC, Shyy JY. Inflammasome Activation in Children With Kawasaki Disease and Multisystem Inflammatory Syndrome. *Arterioscler Thromb Vasc Biol.* 2021 Sep;41(9):2509-2511

We apologise for not including the Wang paper in our previous version of the MS which we have now included in the text. The Papadopoulos paper is a review article rather than a primary research article which references Wang et al., 2021 as its source of information on NLRP3 activation. We have revised the discussion to include the Wang findings as follows:

Our functional assays revealed that canonical NLRP3 activation appears to not be overactive in MIS-C at acute or follow-up stage. This differs from the proposed NLRP3 pathway activation seen in Kawasaki disease where transcriptional changes in the genes associated with NLRP3 pathway were identified⁴⁰. Our findings contrast with a previous study which compared inflammatory profiles of children with Kawasaki disease to samples from three children with MIS-C. In these MIS-C patients increased inflammasome activity was found when measuring caspase 4 and caspase 1 cleavage in granulocytes⁴⁹. Activation of caspase 1 could be downstream of inflammasome activating receptors such as AIM2, PYRIN, NLRC4 and NLRP1, not just NLRP3 (PMID: 35080918). IL-1 β cytokine levels, another readout of inflammasome activity (PMID: 32895525), are not high in children with MIS-C (PMID: 34261329, 34632327, 34912824, 35177862), although this may be covered by treatment effect. The largest randomised trial of immunomodulatory treatments in MIS-C (PIMS-TS, MIS-C; RECOVERY) found neither intravenous immunoglobulin nor anakinra (IL-1 receptor antagonist) had any effect on duration of hospital stay compared with usual care (PMID: 38272046). Examining thus whether inflammasomes are over activated in these patients remains controversial requiring further work in a larger number of patients.

Results

Also in this context, data on Fig.5 are confusing. On page 11, the authors state As expected, addition of LPS plus ATP induced high levels of IL-1 β in the blood of non-MIS-C patients (paediatric admission and COVID-19 pneumonia) at both the 4h and 24h time points (Figure 5A). LPS plus ATP also increased levels of IL-18 in 254 blood samples from the paediatric admission patients, although only at the 24h time point (Figure 5B). However, for MIS-C patients, LPS plus ATP induced a much smaller increase in IL-1 β and IL-18 levels were unchanged.

We have modified the sentences as below to include comments on this group.

As expected, addition of LPS plus ATP induced high levels of IL-1 β in the blood of non-MIS-C patients (paediatric admission and COVID-19 pneumonia) and children recovered from MIS-C at both the 4h and 24h time points (Figure 5A).

The authors, however, do not comment on high levels of IL-1 β in MIS-C follow-up group (Fig.5A)?

In this figure we analysed inflammasome activity of patient monocytes that were either unstimulated or after stimulation with LPS, ATP or LPS plus ATP. The high level is not basal but the level after stimulation. In unstimulated MIS-C follow-up cells IL1 β release was below the level of detection (18pg/ml) indicated by grey shading on the updated figure. Our analysis suggests aberrant inflammasome activity could not be demonstrated, although we cannot rule this out completely because of the limited sensitivity of our assay system.

Figure 5 Lack of overactivation of NLRP3 inflammation activation in MIS-C and increased active caspase 8 activity in MIS-C monocytes. Whole blood samples from children with MIS-C, acute COVID-19 pneumonia (COVID-19, n=15), other paediatric admissions (Paediatric Admission, n=16) and MIS-C follow-up patients (MIS-C_Follow, collected about 1 month post hospital discharge, n=6) were stimulated *in vitro* with LPS (50ng/ml), ATP (5mM), or LPS plus ATP (50ng/ml and 5mM respectively, ATP was added 3h after the LPS addition).

Cytokines were measured 4h and 24h after stimulation. Levels of IL-1 β , IL-18 and TNF- α levels were shown in (A)-(C). The lower limit of quantification of IL-1 β is about 18pg/ml, 16.5pg/ml for IL-18 and 7.3pg/ml for TNF- α (gray shaded areas). In MIS-C group, samples with glucocorticoids/IVIG treatment in the last 24h before sampling were marked as unfilled circle. Horizontal line and error bars show the mean and standard deviation. Results of two-way ANOVA with post hoc Tukey's multiple comparisons test between first 3 groups (all collected at acute stage) within each treatment condition are shown: ** = $p < 0.05$, *** = $p < 0.01$, **** = $p < 0.001$, **** = $p < 0.0001$. There were no significant differences between MIS-C-Follow and paediatric admission group by multiple Mann-Whitney tests with each treatment condition. (D) Monocyte active caspase staining signals from 5 MIS-C and 6 COVID-19 blood monocyte samples. Left: histograms of active caspase 8 staining (red lines indicate MIS-C, blue COVID-19). Right: Quantification of active caspase 8 median fluorescence intensity. Mann-Whitney test was used to compare the 2 groups, **, $p < 0.01$.

Again, is there a possibility that IL-1 β levels in acute MIS-C were suppressed by intensive therapy?

It is possible that IL-1 β could be suppressed by intensive therapy in MIS-C patients. In our discussion, we added this sentence as below. However, in PMID 34632327, the author collected all MIS-C samples before immunosuppression treatment started. We added the point in the discussion about NLRP3 (as below).

IL-1 β cytokine levels, another readout of inflammasome activity (PMID: 32895525), are not high in children with MIS-C (PMID: 34261329, 34632327, 34912824, 35177862), although this may be confounded by treatment effect.

The evidence for lack of activation of NLRP3 in MIS-C patients is also not convincing. Fig.6 F-H, showing ASC, is not very well visible and does not include a proper control – only MIS-C sample in acute and follow-up stages.

We have increased the quality of these images showing magnified field views. The positive control is from a healthy donor blood sample treated with LPS plus ATP. We added control images from COVID-19 blood samples, which shows ASC speck formation (pointed by white arrow).

Representative images from samples where ASC was immunolocalised in whole blood fixed with cytodelic kits including positive control (healthy subject blood samples stimulated with LPS plus ATP to induce NLRP3 activation), MIS-C sample (n=3), MIS-C discharge sample (n=2) and acute COVID-19 sample (n=1). ASC specks can be seen in the positive samples (pointed by white arrow), as well as acute COVID-19 samples, but not MIS-C or MIS-C follow-up samples. Blue staining: DAPI, green staining: ASC.

Considering that Fig.5 shows high levels of IL-1 β in MIS-C follow-up samples, there still might be some activation of NLRP3 ongoing?

We apologise for our lack of clarity in the manuscript text. As explained above, we did not find high basal levels of IL-1 β in MIS-C follow-up samples. In the absence of elevated IL-1 β we were not able to conclude there is activation of NLRP3 ongoing.

Overall, the title, *NLRP3 independent IL-18 signaling orchestrates* seems too strong and supported only indirectly by presented data.

We agree and have changed the title to “IL-18 signalling orchestrates TCR V β 21.3+ T cell activation in acute multisystem inflammatory syndrome in children (MIS-C)”.

Reviewer #3 (Remarks to the Author) from previous reviewer comments:

The authors of this study present a detailed analysis of selected immune parameters in children with MIS-C, compared with children hospitalized for COVID-19 pneumonia and another cohort of children with a severe course of other types of infections. The analysis included a spectrum of serum cytokines, immunophenotyping of T cell subpopulations, complemented by functional studies reflecting NLRP3 inflammasome activation and RNA analysis of selected parameters obtained during the course of the disease.

A fundamental problem with this study arises early in its design in forming patient groups and obtaining samples.

Patient cohorts and cohort groups are insufficiently described. Only basic facts as age, sex and ethnicity are given. The cohort of healthy controls that appear in some experiments is not defined at all. Almost no characteristics on MIS-C clinical variability is presented (although all children were diagnosed according to UK Royal College criteria). Similarly, details are missing for the control group with serious infections that are not specified.

We have added further information about the healthy controls in the supplementary file 1, as well as information about the none COVID-19 related control patients in supplementary table 1. For MIS-C patients, we added information about their clinical variability and a new table detailing the timeframe of treatment and sampling in the same file. We have provided as much detail as we can, but our ethical approval and the need to respect patient confidentiality provides a limit to the granularity of such data.

However, most importantly, samples from children in the MIS-C cohort were all obtained after the children had been treated with high doses of immunosuppressive therapy to manage the acute attack of the disease. This treatment is lifesaving and highly significant, involving corticosteroids and high dosed immunoglobulins, among others. The effect of this therapy may

be absolutely crucial in view of the experimental design of the study. The focus of this work is to monitor NLRP3 inflammasome activity, which (according to published papers) is very likely to be affected by this treatment. Aware of this limitation, the authors surprisingly state in their commentary that the interval between treatment and sampling in their study was shorter than in other published work, but this implies that their samples must have been under the influence of these medications.

New replies: We agree that our data functional assay data on MIS-C acute stage samples could be confounded by pre-treatment. To provide greater clarity, we have done the following - 1) we have clearly labelled samples without prior glucocorticoid/IVIG treatment 2) we have updated the discussion to address that the results may be influenced by prior immunosuppressive treatment 3) The conclusion has been amended to state that we are not able to demonstrate evidence of NLRP3 activation and 4) we have provided data from ASC staining in blood cells from children acutely unwell with MIS-C along with acute respiratory COVID-19 and healthy children.

In the setting of the pandemic, the study was undertaken over a period where treatment guidelines and interventions were evolving. We utilised blood samples taken as early as possible after admission to the study sites but have indicated the therapies that might have been given prior to sampling (at the referring local hospitals) in a transparent manner in the manuscript.

We have found two further manuscripts where LPS stimulation was done in MIS-C whole blood samples. Both papers had similar findings to us and found reduced immune responsiveness of MIS-C blood cells. <https://www.medrxiv.org/content/10.1101/2024.02.02.24301686v1.full>
<https://www.nature.com/articles/s41598-023-43390-6>

We believe it is more meaningful to analyse acutely ill patients as soon as possible, rather than waiting for any potential drug-induced effects to subside, since in this case it is not the acute disease that is being studied. Again, we agree that this also means our functional assay may be influenced by treatments.

Bearing in mind of these concerns, we tried to analyse the effect of pre-treatment on NLRP3 activation from our data and in literature. The finding is a bit surprising: NLRP3 pathway is not effectively suppressed by immune suppression drugs like IVIG or steroid.

In our scRNA-seq study, there are 2 patients whose admission samples were collected before IVIG/steroids treatment, 1 pretreated with IVIG and 2 pretreated with GC. When gene expression in monocytes were compared, neither IL1B, IL18 or NLRP3 were different between GC group and no treatment, or between IVIG group and no treatment although we appreciate this is a very limited number of subjects.

This piece of information is interesting as in previous scRNA-seq study of KD patients, at least IL1B was found to be suppressed by treatment (PMID: 34521850). However, in MIS-C, the effect on IL-1 β is less clear. In MIS-C study of treatment effects, neither IL-1 β or IL-18 levels were affected by IVIG or glucocorticoid treatment (Figure 5, PMC7474869). This is in agreement with our data (Supplementary Figure 1G).

To be transparent about this for reader, in our whole blood functional assay, we now labelled the 3 MIS-C samples without pre-treatment with glucocorticoids/IVIG (revised Figure 5). There is no clear pattern of treatment effect in the data spread in MIS-C group.

Instead of only replying on NLRP3 functional assay, we went further to detect NLRP3 activation marker ASC specks in fixed MIS-C blood samples, there is no evidence of increased ASC speck staining, comparing MIS-C admission to discharge samples, or compared to positive control of LPS plus ATP treated blood samples.

In addition, we compared MIS-C NLRP3 function at follow-up stage when there is little treatment effect influence, to paediatric patient samples. There was no overactive NLRP3 activity in children with MIS-C.

In conclusion the literature on the effects of immune suppression reagents on NLRP3 activation is highly variable, and our data do not show clear evidence of suppression of NLRP3 pathway by the pre-treatments. We have now modified the text to explain the potential complications of immunosuppressive therapy and interpretation of our data, and did more analysis related to NLRP3 pathway in MIS-C.

The assumption that children with MIS-C have reduced NLRP3 inflammasome function is therefore debatable, as it is not possible to distinguish between the effect of MIS-C itself or the effect of immunosuppression. The reduced inflammasome function in MIS-C is summarized in Fig. 5, however, according to the data shown here, the reduction is only evident in IL-1 production detected in stimulated samples, whereas IL-18 secretion is higher in the aforementioned cohort of MIS-C patients. The interpretation of the results seems misleading, the data suggesting rather a switch of NLRP3 inflammasome activation to IL-18 dominance (similar situations are described in the literature).

We agree that 'MIS-C have reduced NLRP3' is debatable and thus we reworded it as 'there is no overactive NLRP3 in MIS-C'.

We thank the reviewer for kindly highlighting high IL-18 levels in MIS-C group. As its levels don't change upon stimulation in MIS-C group, we don't think there is active secretion in MIS-C group in our in vitro experiments and thus no switch to IL-18 dominance.

One possible explanation for this observation includes more stability of IL-18 compared to IL-1beta in vivo. In sepsis, IL-18 remained high from day 1 to day 7, while IL-1beta peaked in day 1 and returned to normal in day 7 (PMCID: PMC7889521).

The authors also looked at the production of IL-18 itself, which appears to be based on many cell types, dominated by monocytes and NK cells in MIS-C. On this point, the authors suggest that these cells produce IL-18 through a noncanonical CD95-mediated process, but they show only circumstantial evidence for this.

We agree that ‘the production of IL-18 itself, which appears to be based on many cell types, dominated by monocytes and NK cells in MIS-C’. So we changed the text as below.

‘While IL1B expression was confined to monocytes, IL18 expression is more ambiguous. Nevertheless, it is clear that CD16- NK cells also express high IL18 in MIS-C, in addition to monocytes ((Figure 6E).’

For monocytes, we did caspase 8 (downstream of CD95 and upstream of IL-18) activity assays and found much higher activity in MIS-C than COVID-19 samples (revised Figure 5C). Also, we did IL-18 cytokine staining in fixed blood cells and found high levels of it in monocytes and CD56+ NK cells in MIS-C (revised Figure 6G).

Thus, the conclusions drawn from the part of the manuscript concerning IL-18 production and reduced inflammasome function in children with MIS-C are somewhat confusing and would require further specification.

To address this point, we carried out more work related to NLRP3 marker ASC speck, monocyte caspase 8 activity assay and intracellular IL-18 cytokine assay.

We modified the statement of reduced NLRP3 activity in MIS-C into ‘there is no evidence of inherently overactive NLRP3 activity’.

The next part of the paper deals with the clonality of T lymphocytes in the MIS-C cohort. This aspect has already been repeatedly described, compared to previous works the authors describe increased expression of ICOS, CD28 and IL18R markers on these cells and direct their appearance to Th1, Th17 and Treg compartments. This is a part of the paper not directly related to the NLRP3 inflammasome and it is questionable whether to combine these parts into one communication.

Given the importance of IL-18 and IFN-gamma signature in MIS-C, we think role of T cell and NK cells (major producer of IFN-gamma) being brought together are a strength of this manuscript. The data on immune cell activation through data on T cell activation is important and as the reviewer points out, this aspect offers novel insights of the T cell markers in MIS-C, and provided detailed T helper cell characterisation. All these are new to understand how superantigen activates T cells in MIS-C. In addition, most previous authors focused on the role of CD8+ TCR Vbeta 21.3+ cells. Our work provided detailed analysis on the contribution of CD4+ cells and found surface markers in CD4 T cells are much better than CD8 T cells for diagnosis of MIS-C.

Overall, this is a communication that complements the already known facts about MIS-C and highlights the potentially more important role of IL-18 in the disease. However, it appears that more detailed evidence would be needed to support its conclusions.

We cannot find evidence that explores the activity of the NLRP3 inflammasome in MIS-C patients before and thus these data are new. In terms of T cells, highlighting the IL-18 link in MIS-C is very helpful to understand the disease mechanism.

Our T cell works are novel. We apologise this is not well introduced in the beginning. So far, there have been no high dimensional studies directly on TCR Vb21.3+ T cells. All previous works only focused on CD8+ T cells and the CD4+ compartment has been overlooked. In our paper, we clearly showed there are a large number of TCR Vb21.3+ CD4+ T cells in MIS-C, co-expressing lots of stimulation markers. It is novel that they not only displayed Th1 but also Treg phenotypes. Our work also shows for the first time how T cell activation in MIS-C compares to T cells in children with sepsis, showing a great similarity. In addition, we found activation markers in TCR Vb21.3+ CD4+ T cells, rather the widely reported CD8+ T cells are characteristic of the inflammatory profile seen at admission in MIS-C.

REVIEWERS' COMMENTS

Reviewer #3 (Remarks to the Author):

The authors have addressed my concerns.